# Reinforcement Learning Fine-Tuning Enhances Activation Intensity and Diversity in the Internal Circuitry of LLMs

**Honglin Zhang,**[*] **Qianyue Hao,**[*] **Fengli Xu, Yong Li**[†]
Department of Electronic Engineering, BNRist, Tsinghua University
Beijing, China

## Abstract

Large language models (LLMs) acquire extensive prior knowledge through large-scale pretraining and can be further enhanced via supervised fine-tuning (SFT) or reinforcement learning (RL)-based post-training. A growing body of evidence has shown that RL fine-tuning improves the capability of LLMs beyond what SFT alone achieves. However, the underlying mechanisms why RL fine-tuning is able to enhance the capability of various LLMs with distinct intrinsic characteristics remain underexplored. In this study, we draw inspiration from prior work on edge attribution patching (EAP) to investigate the internal differences of LLMs before and after RL fine-tuning. Our analysis across multiple model families and mathematical datasets shows two robust effects of online RL post-training: (*i*) an overall increase in average activation intensity, indicating that more internal pathways are engaged and their signals become stronger, and (*ii*) greater diversity in activation patterns, reflected by higher entropy and less concentrated edge distributions. These changes suggest that RL reshapes information flow to be both more redundant and more flexible, which may explain its advantage in mathematical generalization. Notably, models fine-tuned with Direct Preference Optimization (DPO) deviate from these trends, exhibiting substantially weaker or inconsistent internal changes compared to PPO- and GRPO-based training. Together, our findings provide a unified view of how RL fine-tuning systematically alters the internal circuitry of LLMs and highlight the methodological distinctions between online RL and preference-based approaches. Our code is open source at `https://github.com/tsinghua-fib-lab/llm_rl_probing_analysis`.

## 1 Introduction

Recent strides in large language models (LLMs) have shifted the developmental focus from pretraining to post-training (Kumar et al., 2025). A wide array of post-training strategies, ranging from supervised fine-tuning (SFT) (Dong et al., 2023) to reinforcement learning (RL) (Zhang et al., 2025b; Hao et al., 2025), has been developed to enhance model performance. Particularly, RL-based fine-tuning has witnessed rapid advancements, encompassing the development of reward models from Outcome Reward Models (ORM) (Lyu et al., 2025) to Process Reward Models (PRM) (Lightman et al., 2023; Yuan et al., 2024), alongside training algorithms like Proximal Policy Optimization (PPO) (Schulman et al., 2017) and Group Relative Policy Optimization (GRPO) (Shao et al., 2024). With such advancements, emerging empirical evidence indicates that RL-based fine-tuning can enhance the capability of LLMs beyond what is achieved by SFT alone (Chu et al., 2025), improving performance across a range of downstream tasks, including writing (Liao et al., 2025), reasoning (Guo et al., 2025; Xu et al., 2025), and coding (Guo et al., 2024).

Seeking to understand the role of different components within Large Language Models (LLMs) and the origins of their powerful capabilities, a growing body of research has focused on probing their internal structures. Initial studies revealed the working mechanisms of LLMs when solving

---

[1]Honglin Zhang and Qianyue Hao contribute equally to this work.
[2]Yong Li is the corresponding author, email: `liyong07@tsinghua.edu.cn`

mathematical problems by analyzing and statistically examining their internal weights (Shao et al., 2025). Subsequently, some research has analyzed patterns in LLM weights by training external neural probes, which are lightweight auxiliary models (Kim et al., 2025; Zheng et al., 2025). Recently, researchers have investigated the internal residual pathways of LLMs from a graph-theoretic perspective. They have developed methods such as Automated Circuit Discovery (ACDC) (Conmy et al., 2023) and Edge Attribution Patching (EAP) (Syed et al., 2023; Hanna et al., 2024), which assign importance scores to edges or sub-modules and reveal internal functional circuits that determine the capabilities of LLMs.

Despite these advances, existing studies on RL-based post-training have predominantly focused on the external behavioral changes of LLMs, while the underlying internal mechanisms remain under-explored (Ren & Sutherland, 2024). Conversely, works that do investigate the internal mechanisms concentrate on given LLMs, but do not correlate the internal mechanisms to the RL-based post-training methodology with which the LLMs are commonly obtained (Hanna et al., 2024; Kim et al., 2025). As a result, the two lines of research, external evaluation of RL effects and internal mechanistic analysis, have largely progressed in parallel. This gap is partly due to the primary goal of RL post-training, namely enhancing the ability of LLMs to solve complex reasoning tasks, which makes it nontrivial to directly transfer analytical strategies developed on toy problems to the study of RL-induced improvements in real-world problem-solving capabilities.

To address this, we construct a framework for systematically analyzing the mechanisms through which RL fine-tuning affects LLMs. Specifically, we adopt an efficient Edge Attribution Patching (EAP) framework (Nanda, 2023), leveraging the cross-entropy computed from partially truncated generations on mathematical problem-solving tasks to estimate the contribution weights of internal edges. Based on these estimated importance weights, we analyze their distributions before and after RL fine-tuning to interpret changes in internal neuron activations and derive general conclusions regarding the structural effects of RL in the context of mathematical problem solving. Experiments across multiple LLM pairs on diverse mathematical datasets demonstrate that RL post-training strengthens the activation intensity of internal edge connections and diversifies activation patterns during problem-solving. Notably, these effects are not consistently observed under DPO training, highlighting differences between DPO and other RL paradigms, which aligns with prior observations in the literature (Xu et al., 2024).

Overall, the uncovered patterns hold across diverse LLM families, each with distinct characteristics such as architecture and training corpus, suggesting a set of common internal effects induced by RL fine-tuning on reasoning-heavy tasks. These findings provide new insights into how RL post-training reshapes the internal circuitry of LLMs, thereby bridging empirical performance gains with interpretable shifts in internal information pathways. In doing so, they offer guidance for the future development of both LLMs and post-training methodologies.

## 2 PRELIMINARIES

### 2.1 LARGE LANGUAGE MODELS

Large language models (LLMs) are typically built upon the Transformer architecture, comprising a stack of $L$ identical layers (Vaswani et al., 2017; Liu et al., 2024; Bai et al., 2023; Achiam et al., 2023). Each layer consists of two primary sub-structures: a multi-head self-attention mechanism and a position-wise feed-forward network (FFN), each surrounded by a residual connection. The mathematical formulation described below represents the most common architecture found in contemporary LLMs. Let $\mathbf{H}^{(2\ell)} \in \mathbb{R}^{B \times P \times d_{\text{model}}}$ denote the input hidden state to the $(\ell + 1)$-th layer, where $B$ is the batch size, $P$ is the sequence length, and $d_{\text{model}}$ is the hidden dimension. Specifically, the raw input embeddings are denoted by $\mathbf{X}_{\text{input}} = \mathbf{H}^{(0)}$.

The output of the $\ell$-th layer, $\mathbf{H}^{(2\ell)}$, is computed via the sequential processing of the attention and FFN sub-structures. For the attention sub-structure, the input is first normalized as $\mathbf{X}_{\text{attn}}^{\ell} = \text{LayerNorm}\left(\mathbf{H}^{(2\ell-2)}\right)$. The attention mechanism is then applied:

$$\text{Attention}\left(\mathbf{X}_{\text{attn}}^{\ell}\right) = \mathbf{O}_{\text{attn}}^{\ell} = \text{softmax}\left(\frac{\left(\mathbf{X}_{\text{attn}}^{\ell}\mathbf{W}_q^{\ell}\right)\left(\mathbf{X}_{\text{attn}}^{\ell}\mathbf{W}_k^{\ell}\right)^T}{\sqrt{d_k}}\right)\left(\mathbf{X}_{\text{attn}}^{\ell}\mathbf{W}_v^{\ell}\right)\mathbf{W}_o^{\ell}, \quad (1)$$

where $\mathbf{W}_q^\ell, \mathbf{W}_k^\ell \in \mathbb{R}^{d_{\text{model}} \times d_{\text{query}}}, \mathbf{W}_v^\ell \in \mathbb{R}^{d_{\text{model}} \times d_{\text{attn}}}, \mathbf{W}_o^\ell \in \mathbb{R}^{d_{\text{attn}} \times d_{\text{model}}}$ are the query, key, value and output projection matrices, respectively. Here, $d_{\text{query}}$ is the dimensionality of the query and key vectors, and $d_{\text{attn}}$ represents the dimensionality of the value vectors within the attention computation. Positional embeddings are omitted for simplicity. The residual connection yields the intermediate state: $\mathbf{H}^{(2\ell-1)} = \mathbf{H}^{(2\ell-2)} + \mathbf{O}_{\text{attn}}^\ell$. The FFN sub-structure then processes $\mathbf{H}^{(2\ell-1)}$ after normalization: $\mathbf{X}_{\text{ffn}}^\ell = \text{LayerNorm}\left(\mathbf{H}^{(2\ell)}\right)$. The FFN employs a gated mechanism with parallel pathways:

$$\text{FFN}(\mathbf{X}_{\text{ffn}}^\ell) = \mathbf{O}_{\text{ffn}}^\ell = \left(\text{Activation}\left(\mathbf{X}_{\text{ffn}}^\ell \mathbf{W}_{\text{gate}}^\ell\right) \odot \left(\mathbf{X}_{\text{ffn}}^\ell \mathbf{W}_{\text{up}}^\ell\right)\right) \mathbf{W}_{\text{down}}^\ell, \tag{2}$$

where $\mathbf{W}_{\text{gate}}^\ell \in \mathbb{R}^{d_{\text{model}} \times d_{\text{ff}}}, \mathbf{W}_{\text{up}}^\ell \in \mathbb{R}^{d_{\text{model}} \times d_{\text{ff}}}$, and $\mathbf{W}_{\text{down}}^\ell \in \mathbb{R}^{d_{\text{ff}} \times d_{\text{model}}}$ are learned weight matrices, $\odot$ denotes element-wise multiplication, and $d_{\text{ff}}$ is the expanded inner dimension of the FFN. The final output of the layer is obtained via another residual connection: $\mathbf{H}^{(2\ell)} = \mathbf{H}^{(2\ell-1)} + \mathbf{O}_{\text{ffn}}^\ell$. After processing by all $L$ layers, the final hidden states $\mathbf{H}^{(2L)}$ are projected to vocabulary logits via:

$$\mathbf{L} = \mathbf{P}\left(\mathbf{H}^{(2L)}\right) = \text{LayerNorm}\left(\mathbf{H}^{(2L)}\right) \mathbf{W}_{\text{emb}}^T, \tag{3}$$

where $\mathbf{W}_{\text{emb}} \in \mathbb{R}^{V \times d_{\text{model}}}$ is the output embedding matrix and $V$ is the vocabulary size. The resulting tensor $\mathbf{L} \in \mathbb{R}^{B \times P \times V}$ contains the unnormalized logits for each token position.

## 2.2 UNIFIED VIEW OF LLM POST-TRAINING

Previous studies have shown that various post-training methods can be expressed within a unified framework (Shao et al., 2024), encompassing both supervised fine-tuning (SFT) and reinforcement learning (RL)-based approaches. Let $\pi_\theta$ denote the current policy parameterized by $\theta$, and let $(q, o)$ represent a query–response pair. The update rule of a generic post-training algorithm $\mathcal{A}$ can then be written in gradient form as

$$\nabla_\theta \mathcal{J}_\mathcal{A}(\theta) = \mathbb{E}_{(q,o)\sim\mathcal{D}}\left[\frac{1}{|o|}\sum_{t=1}^{|o|} GC_\mathcal{A}(q, o, t, \pi_{\text{rd}}, \pi_{\text{ref}}, \pi_\theta)\nabla_\theta \log \pi_\theta(o_t \mid q, o_{<t})\right], \tag{4}$$

where $\mathcal{D}$ specifies the sampling distribution that generates the training pairs $(q, o)$, $\pi_{\text{rd}}$ denotes the reward model or evaluation rule that produces the learning signal, $\pi_{\text{ref}}$ is the reference policy used to anchor relative preference or advantage computations, and $GC_\mathcal{A}$ represents the token-level weighting factor derived from these signals in algorithm $\mathcal{A}$. This abstraction places different post-training approaches within a unified mathematical representation, enabling direct comparison between supervised and reinforcement-driven update mechanisms.

## 3 METHOD

Our methodology is based on the *Edge Attribution Patching* (EAP) framework (Syed et al., 2023; Hanna et al., 2024; Nanda, 2023), which adopts a graph-theoretic view of LLMs via their residual pathways, reflecting a perspective that has long been present in prior research. While the original work focuses on automated circuit discovery, we adapt its core principle of deriving gradient-based attribution scores for edges to analyze internal information flow differences between models before and after reinforcement learning (RL) fine-tuning.

## 3.1 GRAPH VIEW OF TRANSFORMER RESIDUAL COMPUTATION

Owing to the residual connections in Transformer layers, the input to any sub-module, whether an attention branch or an FFN branch, corresponds to the sum of all preceding sub-module outputs, including the original embedding input. For simplicity, let the attention branch transformation be denoted as $\mathbf{O}_{\text{attn}}^\ell = \mathbf{A}^\ell\left(\mathbf{H}^{(2\ell)}\right)$ and the FFN transformation as $\mathbf{O}_{\text{ffn}}^\ell = \mathbf{F}^\ell\left(\mathbf{H}^{(2\ell+1)}\right)$. Within each attention block, the input is projected to compute queries, keys, and values. These projections can be viewed as three parallel computational paths. For clarity of exposition and ease of presentation, the attention block is treated as a single composite transformation and these internal branches are not explicitly expanded. Then the hidden states satisfy:

$$\mathbf{H}^{(2\ell)} = \mathbf{H}^{(0)} + \sum_{i=1}^{\ell} \mathbf{O}_{\text{attn}}^i + \sum_{j=1}^{\ell} \mathbf{O}_{\text{ffn}}^j, \quad \mathbf{H}^{(2\ell+1)} = \mathbf{H}^{(0)} + \sum_{i=1}^{\ell+1} \mathbf{O}_{\text{attn}}^i + \sum_{j=1}^{\ell} \mathbf{O}_{\text{ffn}}^j. \tag{5}$$

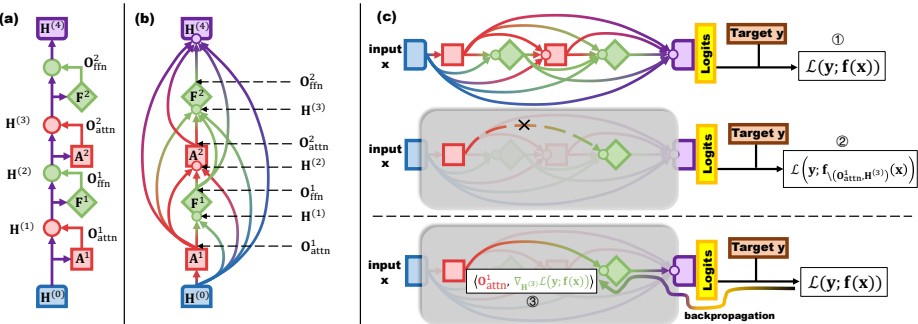

Figure 1: Schematic of a two-layer simplified LLM. (a) Residual perspective, (b) graph perspective, and (c) edge importance estimation: above the dashed line, ACDC-style methods measure the loss change after edge ablation (② − ①), and below, EAP-style methods approximate this via backpropagated gradients (−③ ≈ ② − ①).

Consequently, each sub-module, namely any attention block $\mathbf{A}^\ell$ or feed-forward block $\mathbf{F}^\ell$, can be interpreted as a node in a directed graph. Let us define the set of nodes as

$$\mathcal{V} = \left\{ \mathbf{A}^1, \mathbf{F}^1, \mathbf{A}^2, \mathbf{F}^2, \ldots, \mathbf{A}^L, \mathbf{F}^L \right\}, \tag{6}$$

where $\mathbf{H}^0$ corresponds to the original embedding input. The directed edges, representing the flow of information from sub-module outputs to subsequent inputs, can be formalized as

$$\mathcal{E} = \left\{ \left( \mathbf{H}^{(0)}, \mathbf{H}^{(j)} \right) \mid 1 \le j \le 2L \right\} \cup \left\{ \left( \mathbf{O}_{\text{attn}}^i, \mathbf{H}^{(2\ell-1)} \right), \left( \mathbf{O}_{\text{ffn}}^i, \mathbf{H}^{(2\ell)} \right) \mid 1 \le i \le \ell \le L \right\}. \tag{7}$$

Thus, the LLM can be represented as a directed acyclic graph (DAG) $\mathcal{G} = (\mathcal{V}, \mathcal{E})$, in which nodes correspond to individual sub-modules and edges encode the residual information pathways. This graph-theoretic abstraction facilitates analysis of the model both from a network flow perspective and a circuit-based interpretability standpoint, and for a more intuitive comparison of the residual stream view and the graph view, see Fig. 1(a) and (b).

## 3.2 EDGE-LEVEL ATTRIBUTION

To quantify the importance of individual residual edges, prior work like the *Automated Circuit Discovery* (ACDC) evaluates the change in loss when a given edge is removed (Conmy et al., 2023). Concretely, let $(\mathbf{O}, \mathbf{H}) \in \mathcal{E}$ denote a directed edge from output $\mathbf{O}$ of some sub-module to hidden representation $\mathbf{H}$ at a subsequent stage. ACDC defines the edge importance by the loss perturbation:

$$I_{\text{ACDC}}(\mathbf{O}, \mathbf{H}) = \mathcal{L}\left(\mathbf{y}; \mathbf{f}_{\backslash(\mathbf{O}, \mathbf{H})}(\mathbf{x})\right) - \mathcal{L}\left(\mathbf{y}; \mathbf{f}(\mathbf{x})\right), \tag{8}$$

where $\mathbf{f}(\mathbf{x})$ is the model output under input $\mathbf{x}$, $\mathcal{L}(\mathbf{y}; \cdot)$ denotes the supervised loss relative to target $\mathbf{y}$, and $f_{\backslash(\mathbf{O}, \mathbf{H})}$ represents the model with edge $(\mathbf{O}, \mathbf{H})$ ablated (i.e., setting the corresponding contribution to zero). While conceptually straightforward, this procedure requires two forward passes per edge, rendering it computationally infeasible for large-scale attribution.

By contrast, the EAP framework proposes a gradient-based linearization that estimates the same loss perturbation more efficiently. Specifically, for a given edge $(\mathbf{O}, \mathbf{H})$, consider the ablation $\mathbf{H} \mapsto \mathbf{H} - \mathbf{O}$, which corresponds to removing $\mathbf{O}$'s contribution. A first-order Taylor expansion around $\mathbf{H}$ yields the following compact expression:

$$\Delta\mathcal{L}(\mathbf{O}, \mathbf{H}) \approx - \langle \nabla_{\mathbf{H}} \mathcal{L}(\mathbf{y}; \mathbf{f}(\mathbf{x})), \mathbf{O} \rangle \equiv I_{\text{EAP}}(\mathbf{O}, \mathbf{H}), \tag{9}$$

where $\nabla_{\mathbf{H}} \mathcal{L}(\mathbf{y}; \mathbf{f}(\mathbf{x})) \in \mathbb{R}^{B \times P \times d_{\text{model}}}$ is the loss gradient with respect to the hidden state $\mathbf{H}$, and $\langle \cdot, \cdot \rangle$ denotes the Euclidean inner product.

Considering the computational cost of analyzing large-scale LLMs, we adopt $I_{\text{EAP}}$ to estimate edge-level importance. Importantly, $I_{\text{EAP}}$ can be computed for all edges simultaneously with a single forward and backward pass under the zeroing perturbation, as both the forward activations $\mathbf{O}$ and

the backward gradients $\nabla_{\mathbf{H}}\mathcal{L}$ are available. This approach enables scalable, fine-grained circuit analysis without the need for separate per-edge ablations, making it tractable even for very large models. For a more intuitive comparison of ACDC-style ablation and EAP-style gradient-based attribution, see Fig. 1(c).

### 3.3 SAMPLE SELECTION AND TOKEN-LEVEL TRUNCATION

To ensure fair and tractable edge attribution analysis, we implement a systematic filtering and truncation procedure on model-generated token sequences. Let each model in a paired set generate a token sequence $\mathbf{s}^{\text{base}} = (s_1^{\text{base}}, \ldots, s_{T_{\text{base}}}^{\text{base}})$ and $\mathbf{s}^{\text{RL}} = (s_1^{\text{RL}}, \ldots, s_{T_{\text{RL}}}^{\text{RL}})$ for a given question, where $T_{\text{base}}^q$ and $T_{\text{RL}}^q$ are the respective sequence lengths.

**Question Filtering.** We first select only questions that are correctly answered by both models, and denote the resulting set as $\mathcal{Q}$. To mitigate biases caused by extremely short or long answers, we compute the mean token length across all selected questions for a given model pair and dataset:

$$\bar{T} = \frac{1}{|\mathcal{Q}|} \sum_{q \in \mathcal{Q}} \frac{T_{\text{base}}^q + T_{\text{RL}}^q}{2}. \tag{10}$$

We then define minimum and maximum allowable lengths, $T_{\min} = \beta \bar{T}$, $T_{\max} = \gamma \bar{T}$, and retain only questions satisfying $T_{\min} \leq T_{\text{base}}^q$ and $T_{\text{RL}}^q \leq T_{\max}$.

Finally, to control for comparable sequence lengths between the base and RL models, we require

$$\frac{|T_{\text{base}}^q - T_{\text{RL}}^q|}{(T_{\text{base}}^q + T_{\text{RL}}^q)/2} < \delta, \tag{11}$$

where $\delta \in (0, 1)$ is a balance coefficient. This ensures that the selected questions are comparable in length across both models, minimizing biases in edge importance estimates.

**Token Truncation and Self-Entropy Computation.** For the filtered set of questions, we define a truncation length $T_{\text{cut}} = \alpha \bar{T}$, where $\alpha > 0$ is a scaling coefficient. Only the first $T_{\text{cut}}$ tokens of each sequence are used. Let $\mathbf{L}_t \in \mathbb{R}^V$ denote the model's logit output at token position $t$, and let $\mathbf{s}_{1:T_{\text{cut}}}$ be the sequence of generated tokens truncated to $T_{\text{cut}}$. We compute the self-entropy (cross-entropy of the model with respect to its own output) as

$$\mathcal{L}_{\text{trunc}} = -\frac{1}{T_{\text{cut}}} \sum_{t=1}^{T_{\text{cut}}} \log \frac{\exp(\mathbf{L}_t[s_t])}{\sum_{v=1}^V \exp(\mathbf{L}_t[v])}, \tag{12}$$

where $s_t$ denotes the token actually generated at position $t$ by the model itself.

This ensures that edge importance is computed based on each model's truncated output, maintaining comparability across sequences while avoiding excessive memory usage for overlong generations.

## 4 EXPERIMENT

### 4.1 EXPERIMENTAL SETTINGS

In our experiments, to ensure both reproducibility and the generality of the conclusions, we employed four pairs of open-source large language models (LLMs) of approximately 7B parameters, each consisting of a base model and its counterpart after post-training:

- **Deepseek-Math** (Shao et al., 2024): Both *deepseek-math-7b-instruct* and *deepseek-math-7b-rl* are official DeepSeek models based on the LLaMA-style Transformer. *deepseek-math-7b-instruct* is instruction-tuned on mathematical datasets such as GSM8K, MATH, and MathInstruct, while *deepseek-math-7b-rl* is further trained from it with reinforcement learning on GSM8K and MATH using the Group Relative Policy Optimization (GRPO) algorithm.
- **Mistral** (Chaplot, 2023; Wang et al., 2023): *mistral-7b-sft* is a supervised fine-tuned version of the Mistral-7B model on the MetaMATH dataset, while *math-shepherd-mistral-7b-rl* is further optimized from it using step-by-step Proximal Policy Optimization (PPO) guided by the

Table 1: Comparison of four model pairs (SFT vs. RL) across three datasets, three evaluation metrics, and four hyperparameter settings. Missing values result from GPU memory overflow.

| Dataset | Metric | Scale $\alpha$ | Deepseek-Math SFT | +GRPO | Mistral SFT | +PPO | Distilled-Qwen SFT | +GRPO | Qwen2.5 SFT | +DPO |
|---|---|---|---|---|---|---|---|---|---|---|
| MATH | Act. Intens. ↑ | 0.03 | 2.29e-3 | **2.64e-3** | 9.47e-7 | **3.61e-6** | 6.18e-4 | **6.87e-4** | 1.11e-3 | **1.13e-3** |
| | | 0.1 | 1.10e-3 | **1.31e-3** | 6.76e-4 | **7.71e-4** | 4.51e-4 | **5.59e-4** | **6.95e-4** | 6.90e-4 |
| | | 0.3 | 7.47e-4 | **7.77e-4** | 4.49e-4 | **4.92e-4** | - | - | **4.39e-4** | 4.21e-4 |
| | | 0.5 | 5.64e-4 | **6.02e-4** | 3.58e-4 | **4.05e-4** | - | - | - | - |
| | Info. Complex. ↑ | 0.03 | 1.96e-1 | **2.01e-1** | **3.39e-2** | 1.58e-2 | 1.81e-1 | **2.30e-1** | **2.11e-1** | 1.74e-1 |
| | | 0.1 | 1.72e-1 | **2.47e-1** | 1.41e-1 | **2.09e-1** | 1.11e-1 | **1.96e-1** | **1.60e-1** | 1.34e-1 |
| | | 0.3 | 2.64e-1 | **4.11e-1** | 4.13e-2 | **2.86e-1** | - | - | 1.10e-1 | **1.34e-1** |
| | | 0.5 | 2.71e-1 | **2.93e-1** | 4.52e-2 | **3.22e-1** | - | - | - | - |
| | Dist. Kurt. ↓ | 0.03 | 3.93e+2 | **2.53e+2** | **4.22e+2** | 5.28e+2 | 6.78e+2 | **5.03e+2** | 3.96e+2 | **3.62e+2** |
| | | 0.1 | 3.57e+2 | **2.23e+2** | 4.51e+2 | **3.07e+2** | 1.27e+3 | **9.20e+2** | 5.44e+2 | **4.83e+2** |
| | | 0.3 | 3.11e+2 | **1.89e+2** | 3.35e+2 | **2.65e+2** | - | - | 8.49e+2 | **7.61e+2** |
| | | 0.5 | 3.03e+2 | **1.89e+2** | 2.85e+2 | **2.20e+2** | - | - | - | - |
| College Math | Act. Intens. ↑ | 0.03 | **2.36e-3** | 2.22e-3 | **1.77e-7** | 1.17e-6 | 7.08e-4 | **7.51e-4** | **1.20e-3** | 1.19e-3 |
| | | 0.1 | **1.24e-3** | 1.21e-3 | 8.23e-4 | **9.06e-4** | 5.15e-4 | **5.76e-4** | **8.11e-4** | 8.10e-4 |
| | | 0.3 | **7.61e-4** | 7.57e-4 | 4.92e-4 | **5.32e-4** | - | - | **4.76e-4** | 4.69e-4 |
| | | 0.5 | 5.87e-4 | **5.99e-4** | 3.87e-4 | **4.47e-4** | - | - | **3.71e-4** | 3.53e-4 |
| | Info. Complex. ↑ | 0.03 | 1.45e-1 | **1.96e-1** | **2.51e-2** | 1.14e-2 | 2.13e-1 | **2.35e-1** | 8.01e-2 | **2.17e-1** |
| | | 0.1 | 2.08e-1 | **2.09e-1** | **1.65e-1** | 1.61e-1 | 1.32e-1 | **1.64e-1** | **1.34e-1** | 1.25e-1 |
| | | 0.3 | 2.20e-1 | **2.89e-1** | **3.29e-1** | 2.88e-1 | - | - | **1.23e-1** | 9.95e-2 |
| | | 0.5 | 2.53e-1 | **2.83e-1** | 2.68e-1 | **3.43e-1** | - | - | **1.11e-1** | 1.05e-1 |
| | Dist. Kurt. ↓ | 0.03 | 4.71e+2 | **2.75e+2** | **4.81e+2** | 8.60e+2 | 5.86e+2 | **5.08e+2** | 4.57e+2 | **3.89e+2** |
| | | 0.1 | 3.48e+2 | **2.88e+2** | 3.80e+2 | **2.64e+2** | 1.15e+3 | **8.88e+2** | 5.31e+2 | **4.60e+2** |
| | | 0.3 | 3.31e+2 | **2.19e+2** | 2.77e+2 | **2.08e+2** | - | - | 7.51e+2 | **6.51e+2** |
| | | 0.5 | 3.31e+2 | **2.12e+2** | 2.54e+2 | **2.22e+2** | - | - | 9.15e+2 | **7.48e+2** |
| GSM8K | Act. Intens. ↑ | 0.03 | **3.08e-3** | 2.76e-3 | 4.83e-7 | **1.17e-6** | 1.06e-3 | **1.15e-3** | 2.13e-3 | **2.19e-3** |
| | | 0.1 | 1.43e-3 | **1.50e-3** | 5.90e-4 | **6.59e-4** | 6.71e-4 | **7.72e-4** | **1.13e-3** | 1.13e-3 |
| | | 0.3 | 7.80e-4 | **8.52e-4** | 3.86e-4 | **4.44e-4** | - | - | 6.46e-4 | **6.49e-4** |
| | | 0.5 | 5.76e-4 | **6.52e-4** | 3.01e-4 | **3.60e-4** | - | - | **4.94e-4** | 4.90e-4 |
| | Info. Complex. ↑ | 0.03 | 1.56e-1 | **1.56e-1** | **6.30e-2** | 4.00e-2 | 2.22e-1 | **3.33e-1** | 2.19e-1 | **2.53e-1** |
| | | 0.1 | 1.50e-1 | **2.30e-1** | 8.43e-2 | **1.49e-1** | 1.60e-1 | **2.64e-1** | 1.64e-1 | **1.80e-1** |
| | | 0.3 | 1.71e-1 | **2.27e-1** | 1.48e-1 | **2.09e-1** | - | - | 1.09e-1 | **1.57e-1** |
| | | 0.5 | 1.37e-1 | **3.23e-1** | 1.69e-1 | **2.66e-1** | - | - | 1.14e-1 | **1.28e-1** |
| | Dist. Kurt. ↓ | 0.03 | 4.73e+2 | **3.05e+2** | **2.05e+2** | 2.18e+2 | 3.81e+2 | **3.44e+2** | 4.68e+2 | **3.95e+2** |
| | | 0.1 | 4.57e+2 | **2.79e+2** | 4.21e+2 | **3.07e+2** | 7.66e+2 | **5.60e+2** | 5.22e+2 | **4.53e+2** |
| | | 0.3 | 3.85e+2 | **2.48e+2** | 3.99e+2 | **2.48e+2** | - | - | 7.17e+2 | **5.88e+2** |
| | | 0.5 | 4.02e+2 | **2.49e+2** | 3.16e+2 | **2.18e+2** | - | - | 7.81e+2 | **6.73e+2** |

MATH-SHEPHERD process reward model on GSM8K and MATH, leading to notable gains in mathematical reasoning accuracy.

- **Distilled-Qwen** (Guo et al., 2025; Chen et al., 2025): *DeepSeek-R1-Distill-Qwen-7B* is a Qwen2.5-based model distilled from DeepSeek-R1, trained via supervised distillation to inherit strong reasoning ability.

- **Qwen2.5** (Qwen et al., 2025; Zhang et al., 2025a): *Qwen2.5-7B-SFT* is fine-tuned with supervised learning on the MATH and Numina-Math datasets, while *Qwen2.5-7B-DPO* is derived from that SFT model via iterative Direct Preference Optimization (DPO).

We conducted extensive analyses on three public mathematical benchmarks: *GSM8K*, *MATH*, and *College Math*. More detailed characteristics of the analyzed LLMs and implementation details are provided in the Appendix A and C. Thorough extensive evaluations on multiple benchmarks shown in Appendix D, the post-training generally improves the capability of different LLMs.

## 4.2 METRICS

In our experiments, we quantify differences in LLM behavior before and after reinforcement learning (RL) fine-tuning by analyzing the internal edge-weight matrices obtained from the graph-based attribution procedure. Let $\mathbf{W}^{(k)} \in \mathbb{R}^{n_o \times n_i}$ denote the edge-weight matrix for sample $k$, with

$k = 1, \ldots, n$. Here, $n_o$ denotes the number of output nodes in the graph structure, and $n_i$ denotes the number of input nodes in the graph structure. The collection of all samples forms a tensor $\mathbf{W} \in \mathbb{R}^{n \times n_o \times n_i}$. Based on this input, we define three complementary metrics:

**Activation Intensity (Act.Intens.).**   This metric quantifies the average magnitude of all edge weights across every sample, output, and input, capturing both how many pathways in the LLM are activated and the strength of their activation:

$$\text{Act.Intens.} = \frac{1}{n \, n_o n_i} \sum_{k=1}^{n} \sum_{o=1}^{n_o} \sum_{i=1}^{n_i} \left| W_{oi}^{(k)} \right|. \tag{13}$$

**Information Complexity (Info.Complex.).**   To capture the heterogeneity and unpredictability of edge activations across the entire dataset, we compute a Shannon entropy over the absolute values of all edges from all samples, flattened into a single vector. Let $p_b$ denote the normalized probability of bin $b$ in a histogram of all $|W_{oi}^{(k)}|$ values, with $B$ bins and a small constant $\epsilon$ to prevent $\log 0$:

$$\text{Info.Complex.} = -\sum_{b=1}^{B} p_b \log(p_b + \epsilon). \tag{14}$$

Higher entropy values indicate more complex and less predictable distributions of edge activations, whereas lower values suggest concentrated or more regular patterns. This metric reflects the diversity of active information pathways within the LLM during inference and highlights how RL fine-tuning may alter the overall internal information structure.

**Distribution Kurtosis (Dist.Kurt.).**   To quantify the overall shape and stability of edge-weight distributions, we first compute the kurtosis of each sample's edge-weight matrix and then average across all samples:

$$\text{Dist.Kurt.} = \frac{1}{n} \sum_{k=1}^{n} \left[ \frac{\frac{1}{n_o n_i} \sum_{o,i} \left( W_{oi}^{(k)} - \mu^{(k)} \right)^4}{\left( \frac{1}{n_o n_i} \sum_{o,i} \left( W_{oi}^{(k)} - \mu^{(k)} \right)^2 \right)^2} - 3 \right], \tag{15}$$

where $\mu^{(k)}$ is the mean edge weight of sample $k$. Values approaching zero indicate that individual edge-weight distributions approximate a normal distribution. Conversely, significant positive or negative values reflect leptokurtic (heavy-tailed) or platykurtic (light-tailed) distributions, respectively. This metric serves to evaluate the impact of RL fine-tuning on the tail behavior and outlier characteristics of the overall activation distribution.

## 4.3   Results and Analysis

Our main experimental results are presented in Table 1. We observe that the three model families, Deepseek-Math, Mistral, and Distilled-Qwen, exhibit largely consistent changes in the metrics before and after RL fine-tuning. Specifically, Activation Intensity and Information Complexity tend to increase, while Distribution Kurtosis tends to decrease. Individual exceptions can be seen in some cases for Deepseek-Math and Mistral. However, as the scaling factor $\alpha$ controlling truncation length gradually increases, these exceptions diminish, and the observed patterns become largely consistent, indicating that the phenomenon is relatively robust. However, beyond these observations, we also find several differences in the experimental results of the Qwen2.5 series models trained with the DPO method. For instance, their activation strengths do not exhibit a clear increasing trend, and on the College Math dataset, as $\alpha$ grows to larger values, the Information Complexity metric of their internal pathways remains lower than that of the initial SFT model from which training began.

Taken together, the above observations suggest two key conclusions: **(i)** Online RL fine-tuning for mathematical reasoning increases the extent and intensity of active information edges in the model. **(ii)** Online RL fine-tuning diversifies the activation patterns across these information pathways. We next provide further analyses to substantiate these conclusions.

**Pathway Engagement Induced by RL Fine-tuning.**
As shown in our main results (Table 1), RL fine-tuning consistently increases Act.Intens. and decreases Dist.Kurt., meaning that a substantial number of low-activation edges become more active, effectively engaging a larger set of pathways. This trend is observed across different models, datasets, and hyperparameter settings. Figure 2 illustrates this effect with a representative case: the Mistral model on the MATH dataset at $\alpha = 0.5$. The relative change analysis highlights that many connections strengthen after PPO-based RL fine-tuning, confirming that reinforcement learning systematically enhances the propagation of internal signals.

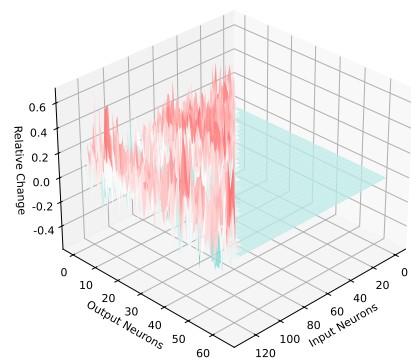

Figure 2: Relative change in edge activation strength after RL fine-tuning for the Mistral model on the MATH dataset with $\alpha = 0.5$.

**Diversity of Activation Patterns in Internal Representations.** In parallel, we find that Info.Complex. generally increases and Dist.Kurt. decreases after RL fine-tuning as shown in Table 1, indicating that activation patterns become more diverse. We provide further visualization results relevant to this conclusion, as illustrated in Figure 3: panel (a) shows that across inference samples, the internal activation structures exhibit greater variability after RL, as quantified by an increase in one minus the mean correlation of edge-weight matrices between sample pairs, and panel (b) further demonstrates that output-edge entropy rises across most model–dataset–hyperparameter combinations. Together, these results indicate that RL enriches the connectivity structure of the internal circuitry, leading to more robust and flexible information flow essential for logical deduction. Furthermore, as shown in Figure 3, panel (a), the Qwen2.5 series models trained with the DPO algorithm also exhibit a certain degree of improvement in diversity. However, the magnitude of this improvement is relatively lower than that of models trained with other online RL methods.

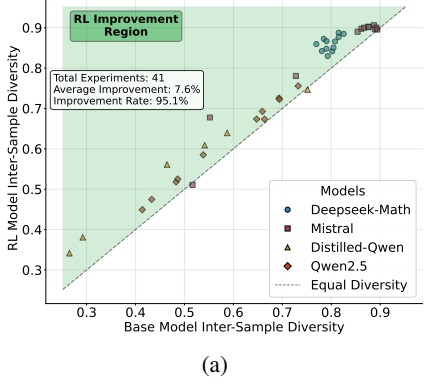

(a)

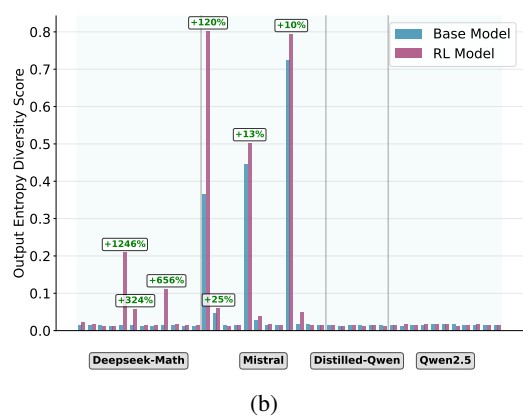

(b)

Figure 3: Comparison before and after RL fine-tuning: (a) diversity of activation patterns across inference samples, including data from all datasets and $\alpha$ values; (b) entropy of output edge patterns per node. In (b), data points are arranged sequentially by dataset (College Math, GSM8K, MATH), iterating over $\alpha \in \{0.03, 0.1, 0.3, 0.5\}$ for each.

Based on Equation (4), we can interpret the observed phenomena by analyzing the fundamental differences in the support of the sampling distribution $\mathcal{D}$ and the properties of the gradient coefficient $GC_{\mathcal{A}}$ across SFT, Online RL, and DPO.

For SFT, the data source is static, drawn from a fixed human-annotated distribution $\mathcal{D}_{SFT} = \{(q, o) \sim P_{\text{data}}\}$, with a constant gradient coefficient $GC_{SFT} = 1$. Consequently, the model optimizes its internal representations to minimize cross-entropy on a narrow, predefined manifold of "correct solutions." This drives the model to converge towards a low-entropy mode that mimics the

training data, resulting in activations concentrated on a small number of outlier edges (high Distribution Kurtosis) and limited engagement of redundant pathways (low Activation Intensity).

In contrast, Online RL algorithms like PPO and GRPO fundamentally alter the data source by introducing on-policy sampling, where outputs are dynamically generated by the evolving policy itself: $\mathcal{D}_{RL} = \left\{ (q, \{o_i\}_{i=1}^{G}) \mid q \sim P_{\text{data}}, o_i \sim \pi_\theta(\cdot|q) \right\}$. This mechanism significantly expands the stochastic support set of the training distribution beyond the SFT subspace, providing the LLM with a richer set of reasoning path samples for each query $q$. Mechanistically, to handle the expanded state space encountered during exploration, the network is compelled to activate and reinforce latent or "dormant" internal circuits that were underutilized during SFT. Furthermore, the gradient coefficient in online RL varies dynamically based on feedback from the reward model or rule. Taking GRPO as an example, $GC_{GRPO} = \hat{A}_{i,t}(q, o, t, \pi_{rd}) + \beta \left( \frac{\pi_{ref}(o_{i,t}|o_{i,<t})}{\pi_\theta(o_{i,t}|o_{i,<t})} - 1 \right)$. To maximize expected reward, the model is driven to mobilize these less active internal circuits to master relatively "harder" problems, as correct responses to such instances typically yield significantly higher gradient coefficients. The observed increase in Activation Intensity and the simultaneous decrease in Distribution Kurtosis reflect this broader utilization of residual pathways. Moreover, as multiple distinct reasoning paths for the same question are reinforced, the entropy of the internal edge weight distribution increases.

Furthermore, from this unified perspective, we can elucidate why DPO exhibits distinct behaviors, particularly its failure to consistently enhance activation intensity and information complexity. Although DPO is mathematically derived from the RL objective, it operates as an offline (or semi-offline, where datasets are refreshed only periodically) algorithm. Its data source remains closer to a relatively more static distribution: $\mathcal{D}_{DPO} = \{(q, o^+, o^-) \sim P_{\text{data}}\}$, rather than the real-time policy $\pi_\theta$. Since DPO restricts optimization to the fixed support set of an offline dataset and effectively retains only two potentially stale contrasting samples for each query $q$, the mechanistic pressure to expand the network's functional capacity through stochastic sampling is significantly weaker. This explains why Activation Intensity and Information Complexity do not show a consistent upward trend compared to the SFT baseline. However, DPO does successfully reduce Distribution Kurtosis. This is because the preference optimization objective is driven by the gradient coefficient $GC_{DPO} = \sigma \left( \beta \log \frac{\pi_\theta(o^-|q)}{\pi_{ref}(o^-|q)} - \beta \log \frac{\pi_\theta(o^+|q)}{\pi_{ref}(o^+|q)} \right)$. This soft margin mechanism relaxes the strict token-matching constraints of SFT, favoring a broader reward maximization landscape and thereby inhibiting the emergence of high-intensity activation edges to some extent, which can be intuitively understood as mitigating rote memorization. Thus, while DPO attenuates the model's reliance on a few high-intensity edges during inference (low kurtosis), it lacks the on-policy exploration dynamics inherent to Online RL, which are essential for driving the systematic enhancement of average internal activation intensity and diversity.

In summary, we posit that the sampling process is the core factor driving the fundamental internal differences between SFT, DPO, and various online-RL paradigms, which consequently leads to disparities in external performance. In Appendix B, we manipulated the sampling dynamics during online-RL by adjusting the training temperature and observed phenomena consistent with our expectations. These findings provide robust empirical support for our hypothesis.

## 5 RELATED WORKS

### 5.1 INTERPRETABILITY OF REINFORCEMENT LEARNING

The inherent opacity of deep reinforcement learning motivates studies on improving their explainability (Qing et al., 2022). Research in explainable RL can be generally categorized into pre-hoc and post-hoc techniques, where the former seeks to build inherently interpretable agents while the latter focuses on analyzing trained agents. Pre-hoc research direction focuses on creating inherently interpretable agents, such as neuro-symbolic systems that represent policies as mathematical expressions (Landajuela et al., 2021; Delfosse et al., 2023), ensuring transparency by design. On contrast, among post-hoc approaches, feature attribution methods are widely applied to generate saliency maps to highlight influential input features (Hao et al., 2022). Besides, another prominent post-hoc paradigm is policy distillation, where the behavior of a complex neural network is distilled into a simpler surrogate model, such as a decision tree, to provide a global summary of the agent's

strategy (Li et al., 2021). Furthermore, counterfactual methods provide an alternative explanatory lens by answering "what if" questions, identifying the minimal state alterations that would have led to a different action (Puri et al., 2019; Huber et al., 2023).

Collectively, these diverse approaches reflect a field moving from reactive explanation of opaque models towards transparent and trustworthy intelligent agents. However, these research mainly focus on lightweight RL agents for conventional decision-making tasks, while it remains unexplored how RL works in the emerging post-training applications, where LLMs are trained as the agent.

## 5.2 INTERPRETABILITY OF LARGE LANGUAGE MODELS

Research into the interpretability of LLMs has largely progressed along two complementary paradigms: mechanistic interpretability and representation interpretability (Singh et al., 2024). Mechanistic interpretability aims to reverse-engineer the patterns learned by a model by analyzing its fundamental components, such as neurons and attention heads, which often employs causal tracing techniques (Gantla, 2025). For instance, one study traced numerical hallucinations to a "Benford's Curse", identifying a statistical bias learned from training data that was internalized by a small subset of feed-forward network (FFN) neurons, and then causally verified this by demonstrating that pruning these specific neurons corrected numerical errors (Shao et al., 2025). In contrast, representation interpretability mainly investigates what information is encoded in the model's internal activation states via external probing models. A prominent line of work in this area uses lightweight probes varying from linear models (Kim et al., 2025) to graph models (Zheng et al., 2025), decoding concepts within the activation space of the model's middle layers. These discovered representations are not merely correlational, but the learned probe weights can be repurposed as "steering vectors" to causally intervene on the activations during generation, thereby controlling the model's output (Kim et al., 2025). While the former paradigm focuses on how a model computes, the latter reveals what knowledge it represents, together offering a more holistic understanding of these complex systems.

While such studies offer valuable perspectives on LLM interpretability, they predominantly focus on analyzing given LLMs without integrating the training methodology with which the LLMs are obtained into the investigation. In particular, it remains unclear how RL, the widely adopted technique in post-training, is able to broadly enhance the capabilities of diverse LLMs with distinct architectural and functional characteristics.

## 6 CONCLUSIONS

We presented a systematic analysis of how reinforcement learning (RL) fine-tuning reshapes the internal circuitry of large language models (LLMs). Using edge attribution patching, we identified two robust effects across multiple model families: stronger average activation intensity and greater diversity in activation patterns. These findings suggest that online RL enhances both the redundancy and flexibility of information flow, which may underlie its superior generalization ability in mathematical domains. In contrast, DPO fine-tuning produced weaker or inconsistent changes, emphasizing the methodological gap between static preference optimization and dynamic online RL. Our results provide a unified mechanistic perspective on RL post-training and offer guidance for the design of future post-training algorithms.

## 7 LIMITATIONS

Our study acknowledges certain limitations that outline important directions for future research. While the consistency of our results suggests potential broader applicability, our empirical validation is currently confined to mathematical reasoning tasks. Verifying whether these internal circuit dynamics hold in domains with open-ended outputs, such as code generation, creative writing, open-ended dialogue and so on, remains a critical subject for future investigation. We also note the limitation regarding model scale, as the significant memory overhead required for granular internal state analysis prevented us from extending our experiments to models larger than 7B parameters.

## ETHICS STATEMENT

We fully use open-source models and datasets in the paper, which involve no problem regarding privacy and copyright. We cite the resources in Section 4.1. This work does not involve human subjects, discrimination, bias, or fairness concerns,

## REPRODUCIBILITY STATEMENT

For Reproducibility, we describe the general experimental settings in Section 4.1, list the implementation details in Appendix C, and provide our anonymously open-sourced code at `https://github.com/tsinghua-fib-lab/llm_rl_probing_analysis`.

## ACKNOWLEDGMENTS

This work was supported in part by the National Key Research and Development Program of China under 2024YFC3307605, the National Natural Science Foundation of China under 62472241, and National Research Center for Information Science and Technology.

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

# A    CHARACTERISTICS OF ANALYZED LLMS

We employed four pairs of large language models (LLMs), each consisting of a base model (SFT) and its post-trained RL counterpart. The models and their download links are listed below:

- **DeepSeek-Math**
    - deepseek-math-7b-instruct: `https://huggingface.co/deepseek-ai/deepseek-math-7b-instruct`
    - deepseek-math-7b-rl: `https://huggingface.co/deepseek-ai/deepseek-math-7b-rl`
- **Mistral**
    - mistral-7b-sft: `https://huggingface.co/peiyi9979/mistral-7b-sft`
    - math-shepherd-mistral-7b-rl: `https://huggingface.co/peiyi9979/math-shepherd-mistral-7b-rl`
- **Distilled-Qwen**
    - DeepSeek-R1-Distill-Qwen-7B: `https://huggingface.co/deepseek-ai/DeepSeek-R1-Distill-Qwen-7B`
    - AceReason-Nemotron-7B: `https://huggingface.co/nvidia/AceReason-Nemotron-7B`
- **Qwen2.5**
    - Qwen2.5-7B-SFT: `https://huggingface.co/RLHFlow/Qwen2.5-7B-SFT`
    - Qwen2.5-7B-DPO: `https://huggingface.co/RLHFlow/Qwen2.5-7B-DPO`

As summarized in Table 2, these LLMs are designed with distinctive structural and functional characteristics.

Table 2: Structural and functional characteristics of the analyzed LLMs.

| LLM series | Parameter size | # layers | # heads | Max ctx | Dim | Vocab size |
|---|---|---|---|---|---|---|
| DeepSeek-Math | 7B | 30 | 32 | 4096 | 4096 | 102400 |
| Mistral | 7B | 32 | 32 | 4096 | 4096 | 32000 |
| Distilled-Qwen | 7B | 28 | 28 | 131072 | 3584 | 152064 |
| Qwen-2.5 | 7B | 28 | 28 | 8192 | 3584 | 151665 |

# B  TRAINING DYNAMICS UNDER SAMPLING INTERVENTIONS

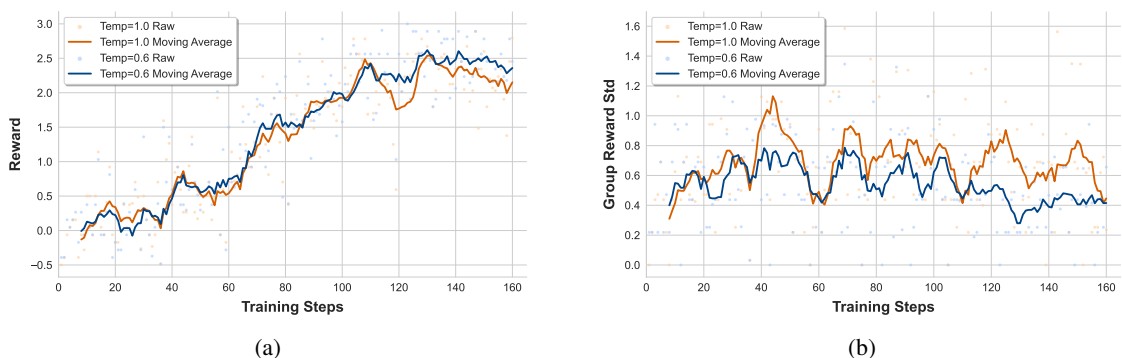

Figure 4: Smoothed moving-average curves of reward and group reward standard deviation during RL training under different temperatures, with a sliding window length of 8.

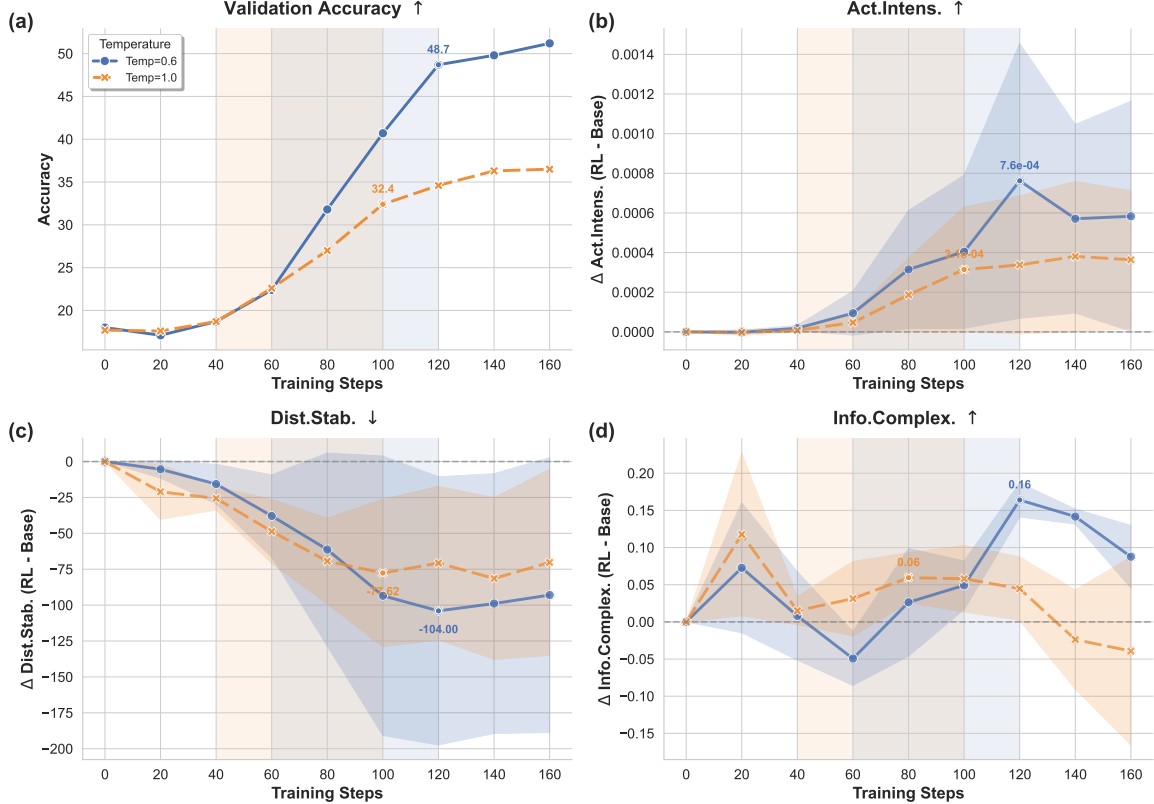

Figure 5: (a) Accuracy of models trained with different numbers of RL steps on the GSM8K test set. (b), (c), and (d) show, respectively, the differences in Activation Intensity, Distribution Stability, and Information Complexity between models trained with different numbers of RL steps and the initial model. We highlight the training intervals where the performance gains are most pronounced for temperature = 0.6 and temperature = 1.0, corresponding to [60, 120] and [40, 100], and mark the extrema of each metric within these intervals in the expected direction.

To investigate whether the observed internal circuit changes are causal drivers of performance improvement rather than mere byproducts, we designed an intervention experiment. These changes specifically refer to the increased activation intensity and the enhancement of pattern diversity. We

employed Qwen2.5-3B-Instruct[1] as the base model and limited its maximum generation length to 200 tokens to constrain its initial reasoning capabilities. On this basis, we conducted reinforcement learning training on the GSM8K training set using an improved variant of the GRPO algorithm (Yu et al., 2025). The experiment was configured with a batch size of 32 and sampled 4 candidate responses per query. We utilized the RLVR reward function where correct answers received 3 points and completely incorrect answers received -0.5 points. Partial correctness was scored proportionally based on the deviation between the prediction and the ground truth.

Our core hypothesis asserts that effective on-policy exploration drives the activation and consolidation of beneficial internal circuits and subsequently leads to performance gains. To verify this, we required a control variable to modulate exploration quality. We selected sampling temperature as the intervention method because excessively high temperatures theoretically introduce excessive randomness and alter sampling effectiveness. We compared two temperature settings. A temperature of 0.6 represents effective exploration under standard settings while 1.0 represents noisy exploration under high-entropy settings.

Figure 4 illustrates the reward trajectories and the standard deviation of the Group Reward during training. As shown in Figure 4(b), the Group Reward standard deviation under the 1.0 temperature setting is significantly higher than that of the 0.6 setting. This confirms that high temperatures induce higher output mode variability (Liu et al., 2025; Ren & Sutherland, 2024). Such excessive variance implies that the signals generated during exploration are noisier. Consequently, it becomes difficult for the model to reliably identify the underlying patterns corresponding to correct solutions.

We tracked the evolution of three key internal metrics relative to the initial SFT model as detailed in Figure 5(b)-(d). We observed significant differences during the critical mid-training phase between steps 40 and 120. At temperature 0.6, activation intensity showed a clear increasing trend and peaked around step 120. In contrast, the growth of this metric was significantly suppressed at temperature 1.0 where the peak was markedly lower than that of the low-temperature group. This suggests that noisy sampling hindered the full activation of latent dormant circuits. Similarly, the information complexity for the 0.6 group rose sharply between steps 100 and 120. Conversely, the 1.0 group failed to sustain growth and even exhibited a decline during mid-training. Regarding distribution kurtosis in Figure 5(c), the 0.6 group showed a significant decrease. This implies that the model no longer relies excessively on a few concentrated high-activation pathways for reasoning. In comparison, the 1.0 group exhibited a smaller decrease. Overall, the 0.6 temperature group achieved higher peaks in activation intensity and information complexity alongside lower valleys in distribution kurtosis compared to the 1.0 group during the mid-training phase.

The differences in internal circuits mapped directly to downstream task performance. Figure 5(a) illustrates the changes in GSM8K test accuracy. The accuracy of the 0.6 group climbed rapidly alongside the expected changes in internal metrics and finally reached 51.2%. Conversely, the performance growth of the 1.0 group was slow due to suppressed internal circuit evolution and eventually stagnated around 36.5%. This is far below the control group. The results indicate that performance gains are substantially weakened when the normal evolution of internal circuits is artificially suppressed via high-temperature sampling. This intervention-suppression effect provides effective empirical support for our causal hypothesis. This provides empirical support for the view that diverse and high-intensity internal pathway activation is a key mechanism bridging effective sampling and performance improvement, rather than being a simple byproduct.

---

[1]https://huggingface.co/unsloth/Qwen2.5-3B-Instruct

## C    IMPLEMENTATION DETAILS

In this section, we provide all implementation details for reproducibility in Table 3. In the experiments, we selected the hyperparameter configuration as $\beta = 0.5$, $\gamma = 1.5$, and $\delta = 0.5$. For each combination of dataset and $\alpha$, we randomly drew 100 pairs of contrastive samples from the final pool of filtered samples.

Table 3: Implementation details

| Module | Element | Detail |
|--------|---------|--------|
|        | OS      | Ubuntu 22.04.3 LTS |
|        | CUDA    | 12.2 |
| System | Python  | 3.11 |
|        | Pytorch | 2.7.0+cu26 |
|        | Device  | 2*NVIDIA A100 80G |

## D PERFORMANCE OF LLMs

Here we compare the performance of LLMs before and after post-training on multiple benchmarks. As shown in Table 4, post-training generally improves the capability of different LLMs.

Table 4: Performance comparisons of LLMs before and after post-training. Bold numbers indicate better performance.

| LLM series | Post-training | MATH | GSM8K | Minerva math | Olympiad bench | College math | AIME24 | AMC23 |
|---|---|---|---|---|---|---|---|---|
| DeepSeek-Math | Before | 46.2 | 82.1 | 22.1 | 14.5 | 30.8 | 3.3 | 17.5 |
| | After | **52.6** | **87.9** | **27.2** | **18.2** | **33.5** | **6.7** | **25.0** |
| Mistral | Before | 29.1 | 78.2 | **12.1** | 5.5 | 17.5 | 0.0 | 12.5 |
| | After | **32.6** | **84.2** | 11.8 | **9.2** | **19.9** | 0.0 | 12.5 |
| DS-Distill-Qwen | Before | 88.4 | 90.3 | 43.0 | 49.8 | 40.0 | 46.7 | 87.5 |
| | After | **95.4** | **93.4** | **55.9** | **65.9** | **44.6** | **70.0** | **95.0** |
| Qwen-2.5 | Before | 75.7 | **92.2** | 32.7 | 37.6 | 41.9 | 16.7 | 62.5 |
| | After | **82.6** | 92.0 | **40.1** | **46.4** | **42.5** | **26.7** | **67.5** |

# E    USE OF LLMS

The authors used LLMs to aid or polish paper writing, but all content has been carefully reviewed by the author. The authors used LLMs for literature retrieval and discovery, but all related works have been carefully reviewed and organized by the author. The research ideation in this work was entirely completed by the author and does not involve the use of LLMs.

