# OpenReview forum: "Reinforcement Learning Fine-Tuning Enhances Activation Intensity and Diversity in the Internal Circuitry of LLMs"
_ICLR.cc/2026/Conference — ICLR 2026 Poster_

### Official Review · Reviewer_kREr · 2025-11-01

**Soundness:** 3
**Presentation:** 3
**Contribution:** 3
**Rating:** 8
**Confidence:** 3

**Summary:**

This paper investigates the underlying mechanisms by which RL fine-tuning enhances the capabilities of LLMs. Leveraging a graph-theoretic perspective and edge attribution patching (EAP), the authors analyze the internal information flow of LLMs by comparing models before (after supervised fine-tuning) and after RL-based post-training. The study finds that online RL algorithms (like PPO and GRPO) systematically alter the internal circuitry of LLMs in two key ways: Increased Activation Intensity (More internal pathways become active, and their signal strengths increase, suggesting a more engaged and robust information flow) and Increased Activation Diversity (Activation patterns become more varied and less concentrated, indicating that the model develops more flexible and diverse pathways for problem-solving). Overall, the work provides a unified, mechanistic explanation for the performance gains seen from RL fine-tuning, bridging the gap between external model behavior and internal circuit-level transformations.

**Strengths:**

S1: The paper’s detailed comparison between online reinforcement learning methods (such as PPO and GRPO) and preference-based techniques like DPO—especially regarding the distinct internal transformations they trigger—offers a new lens through which to understand current LLM fine-tuning paradigms.

S2: The main contributions and results of the study are presented with clarity. The notions of activation intensity and activation diversity are both intuitive and insightful, effectively summarizing the internal changes observed. The discussion linking these changes to improved robustness and adaptability in information processing is straightforward, helping demystify the complex inner workings of LLMs. This lucidity allows the paper’s substantial technical insights to be easily grasped and appreciated by a wide range of AI researchers.

**Weaknesses:**

W1: Although the paper effectively highlights what changes occur—namely, increased activation intensity and diversity—it does not adequately explain why these specific changes arise from online RL. The discussion mainly points to correlations between RL fine-tuning and the resulting internal shifts. To advance beyond correlation, the work should develop clear, testable hypotheses detailing the causal mechanisms by which RL signals (such as reward functions or policy gradient updates) drive alterations in edge attribution and activation dynamics.

W2:  The study focuses on 7B parameter models and primarily LLaMA-style Transformers. While this is a common scale, the generalizability of the findings to a broader spectrum of LLMs remains unclear.

**Questions:**

Q1: For W1, could you share any thoughts on the possible causal factors or how you plan to investigate them?

---

> ### Author Response · Authors · 2025-11-27
> **Rebuttal (Part 1/4)**
>
> We sincerely thank you for your recognition of our work and for your thorough and insightful review. Your valuable comments have been instrumental in helping us further improve the manuscript. Below, we provide a point-by-point response to each of your suggestions and questions.
>
> ## W1
>
> We appreciate your valuable suggestion. In fact, we provided a brief discussion of the underlying causal mechanisms in Section 4.3, “Results and Analysis”. However, due to space constraints, this discussion was not fully elaborated. We have now further expanded and refined this section.  You can refer to **Section 4.3 in the revised PDF**, and for your convenience, we have also included the corresponding content here.
>
>
> > > The gradient update of any post-training method can be expressed in a unified form as follows (Equation (4)) [1]:
> > >
> > > $\nabla_{\theta} \mathcal{J}_\mathcal{A}(\theta) = \mathbb{E}\_{{(q,o)\sim\mathcal{D}}}[ \frac{1}{|o|}\sum\_{t=1}^{|o|} GC\_\mathcal{A}(q,o,t,\pi\_{\text{rd}},\pi\_{\text{ref}},\pi\_\theta)\nabla\_{\theta}\log \pi\_{\theta}(o\_t \mid q,o\_{<t}) ]$
> > >
> > > In this equation, $\mathcal{D}$ denotes the data source, $GC_{\mathcal{A}}$ represents the gradient coefficient derived from reward signals or evaluation rules, $\pi_{\mathrm{rd}}$ denotes the reward model, $\pi_{\mathrm{ref}}$ denotes the reference policy, $\pi_\theta$ denotes the model being trained, $q$ is the query, $o$ is the response generated by the LLM, and $t$ indexes the token position.
> >
> > Based on Equation (4), we can interpret the observed phenomena by analyzing the fundamental differences in the support of the sampling distribution $\mathcal{D}$ and the properties of the gradient coefficient $GC_{\mathcal{A}}$ across SFT, Online RL, and DPO.
> >
> > For SFT, the data source is static, drawn from a fixed human-annotated distribution $\mathcal{D}\_{SFT} = \\{(q, o) \sim P\_{\text{data}}\\}$, with a constant gradient coefficient $GC\_{SFT}=1$. Consequently, the model optimizes its internal representations to minimize cross-entropy on a narrow, predefined manifold of "correct solutions." This drives the model to converge towards a low-entropy mode that mimics the training data, resulting in activations concentrated on a small number of outlier edges (high Distribution Kurtosis) and limited engagement of redundant pathways (low Activation Intensity).
> >
> > In contrast, Online RL algorithms like PPO and GRPO fundamentally alter the data source by introducing on-policy sampling, where outputs are dynamically generated by the evolving policy itself: $\mathcal{D}\_{RL} = \\{(q, \\{o_i\\}\_{i=1}^G) \mid q \sim P\_{\text{data}}, o\_i \sim \pi_\theta(\cdot\mid q) \\}$. This mechanism significantly expands the stochastic support set of the training distribution beyond the SFT subspace, providing the LLM with a richer set of reasoning path samples for each query $q$. Mechanistically, to handle the expanded state space encountered during exploration, the network is compelled to activate and reinforce latent or "dormant" internal circuits that were underutilized during SFT. Furthermore, the gradient coefficient in online RL varies dynamically based on feedback from the reward model or rule. Taking GRPO as an example, $GC_{GRPO}=\hat A_{i,t}(q,o,t,\pi_{rd})+\beta\left(\frac{\pi_{ref}(o_{i,t}|o_{i,<t})}{\pi_\theta(o_{i,t}|o_{i,<t})}-1\right)$. To maximize expected reward, the model is driven to mobilize these less active internal circuits to master relatively "harder" problems, as correct responses to such instances typically yield significantly higher gradient coefficients. The observed increase in Activation Intensity and the simultaneous decrease in Distribution Kurtosis reflect this broader utilization of residual pathways. Moreover, as multiple distinct reasoning paths for the same question are reinforced, the entropy of the internal edge weight distribution increases.
> >

---

> ### Author Response · Authors · 2025-11-27
> **Rebuttal (Part 2/4)**
>
> > Furthermore, from this unified perspective, we can elucidate why DPO exhibits distinct behaviors, particularly its failure to consistently enhance activation intensity and information complexity. Although DPO is mathematically derived from the RL objective, it operates as an offline (or semi-offline, where datasets are refreshed only periodically) algorithm. Its data source remains closer to a relatively more static distribution: $\mathcal{D}\_{DPO} = \\{(q, o^+, o^-) \sim P\_{\text{data}}\\}$, rather than the real-time policy $\pi\_\theta$. Since DPO restricts optimization to the fixed support set of an offline dataset and effectively retains only two potentially stale contrasting samples for each query $q$, the mechanistic pressure to expand the network's functional capacity through stochastic sampling is significantly weaker. This explains why Activation Intensity and Information Complexity do not show a consistent upward trend compared to the SFT baseline. However, DPO does successfully reduce Distribution Kurtosis. This is because the preference optimization objective is driven by the gradient coefficient $GC_{DPO} = \sigma\left(\beta \log \frac{\pi_\theta(o^-|q)}{\pi_{ref}(o^-|q)} - \beta \log \frac{\pi_\theta(o^+|q)}{\pi_{ref}(o^+|q)}\right)$. This soft margin mechanism relaxes the strict token-matching constraints of SFT, favoring a broader reward maximization landscape and thereby inhibiting the emergence of high-intensity activation edges to some extent, which can be intuitively understood as mitigating rote memorization. Thus, while DPO attenuates the model's reliance on a few high-intensity edges during inference (low kurtosis), it lacks the on-policy exploration dynamics inherent to Online RL, which are essential for driving the systematic enhancement of average internal activation intensity and diversity.
>
> In summary, consistent with widely observed empirical consensus, one of the primary factors underlying the differences between various training methods is whether the support set used for training consists of online-sampled data and whether the sampling is sufficiently diverse [2]. In other words, we hypothesize that the phenomena we observed likely arise along the following causal chain:
> $\text{dynamic and informative sampling} \\rightarrow \text{activation of more and richer beneficial internal pathways} \\rightarrow$
>
> $\text{improved model performance}.$ Our additional experiments offer supporting evidence that, to some extent, validates this hypothesis.
>
> To empirically test the hypothesis that sampling dynamics drive internal circuit evolution, which in turn fuels performance, we designed an training experiment using Qwen2.5-3B-Instruct [3]. We constrained the model's maximum generation length to 200 tokens to limit its initial reasoning capability. On this basis, we performed RL training on the GSM8K dataset using an improved variant of the GRPO algorithm [4] with a batch size of 32 and 4 rollout samples per query. We utilized a verifiable reward function (RLVR), assigning +3 for correct answers, -0.5 for incorrect ones, and proportional scores for partial correctness.
>
> We identified sampling temperature as a critical control variable to modulate the quality of the on-policy data distribution $\mathcal{D}_{RL}$. By contrasting a standard exploration setting (temperature=0.6) with a high-temperature setting (temperature=1.0), we aimed to examine whether altering the sampling process would inhibit the emergence of the observed internal patterns, and consequently, suppress performance gains.
>
> We tracked the trajectories of the three key internal metrics relative to the initial SFT baseline across the training process, averaging the differences over $\alpha \in \\{0.1, 0.3, 0.5\\}$. **For a comprehensive visualization of these dynamics, we refer the reviewer to Appendix B in the updated PDF.** Below, we highlight the peak deviations in the expected direction during the critical mid-training phase. Values indicating stronger progression in the expected direction are shown in bold, while the corresponding metrics in the comparison setting are italicized.
>
>
> ### Activation Intensity ($\uparrow$)
>
> | training steps  | initial |   20    |   40    |   60    |   80    |   100   |    120     |   140    |   160   |
> | :-------------: | :-----: | :-----: | :-----: | :-----: | :-----: | :-----: | :--------: | :------: | :-----: |
> | temperature=0.6 |    0    | -2.2e-6 | 1.9e-5  | 9.5e-5  | 3.1e-4  | 4.0e-4  | **7.6e-4** |  5.7e-4  | 5.8e-4  |
> |    $\Delta$     |         | -2.2e-6 | +2.1e-5 | +7.6e-5 | +2.2e-4 | +8.9e-5 |  +3.6e-4   | -1.9e-4  | +1.2e-5 |
> | temperature=1.0 |    0    | -2.4e-6 | 7.5e-6  | 4.7e-5  | 1.9e-4  | 3.1e-4  |   3.4e-4   | *3.8e-4* | 3.6e-4  |
> |    $\Delta$     |         | -2.4e-6 | +9.9e-6 | +4.0e-5 | +1.4e-4 | +1.3e-4 |  +2.3e-5   | +4.3e-5  | -1.7e-5 |

---

> ### Author Response · Authors · 2025-11-27
> **Rebuttal (Part 3/4)**
>
> ### Distribution Kurtosis ($\downarrow$)
>
> | training steps  | initial |   20    |   40    |   60    |   80    |   100   |     120     |    140    |   160   |
> | :-------------: | :-----: | :-----: | :-----: | :-----: | :-----: | :-----: | :---------: | :-------: | :-----: |
> | temperature=0.6 |    0    | -5.4e+0 | -1.6e+1 | -3.8e+1 | -6.1e+1 | -9.3e+1 | **-1.0e+2** |  -9.9e+1  | -9.3e+1 |
> |    $\Delta$     |         | -5.4e+0 | -1.0e+1 | -2.2e+1 | -2.3e+1 | -3.2e+1 |   -1.1e+1   |  +5.1e+0  | +5.9e+0 |
> | temperature=1.0 |    0    | -2.1e+1 | -2.6e+1 | -4.9e+1 | -6.9e+1 | -7.8e+1 |   -7.1e+1   | *-8.1e+1* | -7.0e+1 |
> |    $\Delta$     |         | -2.1e+1 | -4.6e+0 | -2.3e+1 | -2.1e+1 | -8.2e+0 |   +6.9e+0   |  -1.1e+1  | +1.1e+1 |
>
> ### Information Complexity ($\uparrow$)
>
> | training steps  | initial |   20    |   40    |   60    |    80    |   100   |    120     |   140   |   160   |
> | :-------------: | :-----: | :-----: | :-----: | :-----: | :------: | :-----: | :--------: | :-----: | :-----: |
> | temperature=0.6 |    0    | 7.3e-2  | 8.0e-3  | -4.9e-2 |  2.6e-2  | 4.9e-2  | **1.6e-1** | 1.4e-1  | 8.8e-2  |
> |    $\Delta$     |         | +7.3e-2 | -6.5e-2 | -5.7e-2 | +7.6e-2  | +2.3e-2 |  +1.1e-1   | -2.2e-2 | -5.4e-2 |
> | temperature=1.0 |    0    | 1.2e-1  | 1.5e-2  | 3.1e-2  | *5.9e-2* | 5.8e-2  |   4.5e-2   | -2.4e-2 | -3.9e-2 |
> |    $\Delta$     |         | +1.2e-1 | -1.0e-1 | +1.6e-2 | +2.8e-2  | -1.4e-3 |  -1.3e-2   | -6.8e-2 | -1.6e-2 |
>
> The results reveal a clear mechanistic divergence driven by sampling differences. In the standard setting (temperature=0.6), the model exhibits a robust evolution in internal circuitry: Activation Intensity and Information Complexity surge while Distribution Kurtosis drops significantly during the critical learning phase (steps 60–120). In contrast, the high-temperature setting (temperature=1.0) introduces excessive stochasticity into the rollout generation [5] [6]. This noise makes it difficult for the optimizer to reliably identify and reinforce the correct reasoning patterns (as evidenced by the higher group-reward std curve, Figure 4(b) in updated Appendix B). Mechanistically, this "noisy sampling" effectively blocks the consolidation of stable internal pathways: the peaks of Activation Intensity and Information Complexity (during mid-to-late stage) are markedly suppressed, while the minimum of Distribution Kurtosis remains significantly higher. This demonstrates that efficient on-policy sampling is the direct cause of the observed expansion in internal circuits.
>
> To confirm that this internal suppression leads to inferior outcomes, we evaluated the GSM8K test accuracy.
>
> ### Accuracy ($\uparrow$)
>
> | training steps  | initial |  20  |  40  |  60  |  80  | 100  | 120  | 140  |   160    |
> | :-------------: | :-----: | :--: | :--: | :--: | :--: | :--: | :--: | :--: | :------: |
> | temperature=0.6 |  18.0   | 17.1 | 18.7 | 22.4 | 31.8 | 40.7 | 48.7 | 49.8 | **51.2** |
> |    $\Delta$     |         | -0.9 | +1.6 | +3.7 | +9.4 | +8.9 | +8.0 | +1.1 |   +1.4   |
> | temperature=1.0 |  17.7   | 17.6 | 18.7 | 22.6 | 27.0 | 32.4 | 34.6 | 36.3 |  *36.5*  |
> |    $\Delta$     |         | -0.1 | +1.1 | +3.9 | +4.4 | +5.4 | +2.2 | +1.7 |   +0.2   |
>
> The performance results are consistent with our hypothesized causal mechanism. The model trained with temperature=0.6, which successfully developed the favorable internal patterns, achieves significantly higher accuracy. Conversely, the model trained with temperature=1.0, whose internal circuit evolution was inhibited by noisy sampling, suffers from inferior performance. The experiments described above provide empirical support for the potential causal chain we hypothesized.
>
>
>
> [1] Shao Z, Wang P, Zhu Q, et al. Deepseekmath: Pushing the limits of mathematical reasoning in open language models[J]. arXiv preprint arXiv:2402.03300, 2024.
>
> [2] Sachdeva N, Coleman B, Kang W C, et al. How to train data-efficient llms[J]. arXiv preprint arXiv:2402.09668, 2024.
>
> [3] https://huggingface.co/unsloth/Qwen2.5-3B-Instruct.
>
> [4] Yu Q, Zhang Z, Zhu R, et al. Dapo: An open-source llm reinforcement learning system at scale[J]. arXiv preprint arXiv:2503.14476, 2025.
>
> [5] Liu J, Gao F, Wei B, et al. What can rl bring to vla generalization? an empirical study[J]. arXiv preprint arXiv:2505.19789, 2025.
>
> [6] Renze M. The effect of sampling temperature on problem solving in large language models[C]//Findings of the association for computational linguistics: EMNLP 2024. 2024: 7346-7356.

---

> ### Author Response · Authors · 2025-11-27
> **Rebuttal (Part 4/4)**
>
> ## W2
>
> We agree with the reviewer that validating our results on different model sizes is valuable. We primarily used 7B models because of hardware constraints. The detailed analysis of internal states consumes much more memory than the model itself, so 7B is the maximum size our current resources can support. Nevertheless, the 7B scale is a common standard in research [1] [2] [3] and has been proven effective for mathematical reasoning [4].
>
> Regarding smaller models, please see the additional experiments in **our response to W1**. We found that the 3B model follows the same trends during training, which suggests our findings apply to smaller scales as well. We acknowledge that we could not test larger models and have clarified this in the Limitations section of the revised paper.
>
> Regarding model architecture, we tried to cover a diverse range of open-source models. However, since most current models follow the "LLaMA-style" structure [5], it is difficult to find suitable opensource models with significantly different architectures for this analysis. On the other hand, this dominance means that our study covers the most representative model families used today. We have also noted this scope in the Limitations section to ensure rigor.
>
> [1] Zhang Y, Li B, Giannakis G B. RefLoRA: Refactored Low-Rank Adaptation for Efficient Fine-Tuning of Large Models[J]. arXiv preprint arXiv:2505.18877, 2025.
>
> [2] An S, Ma Z, Lin Z, et al. Make your llm fully utilize the context[J]. Advances in Neural Information Processing Systems, 2024, 37: 62160-62188.
>
> [3] Feucht S, Atkinson D, Wallace B C, et al. Token erasure as a footprint of implicit vocabulary items in LLMs[C]//Proceedings of the 2024 Conference on Empirical Methods in Natural Language Processing. 2024: 9727-9739.
>
> [4] Li C, Wang W, Hu J, et al. Common 7b language models already possess strong math capabilities[J]. arXiv preprint arXiv:2403.04706, 2024.
>
> [5] Minaee S, Mikolov T, Nikzad N, et al. Large language models: A survey[J]. arXiv preprint arXiv:2402.06196, 2024.
>
>
> ----
>
> ## Q1
>
> You may refer to **our response to W1**, and we would be very happy to address any further questions you may have.

---

### Official Review · Reviewer_VkAD · 2025-11-01

**Soundness:** 3
**Presentation:** 3
**Contribution:** 2
**Rating:** 4
**Confidence:** 4

**Summary:**

This paper investigates the internal effects of reinforcement learning (RL) fine-tuning on large language models (LLMs), using the Edge Attribution Patching (EAP) framework to analyze residual pathways before and after RL fine-tuning. The authors report two consistent effects across several model families: (1) increased activation intensity and (2) greater diversity of activation patterns. They argue that these internal changes help explain the superior generalization performance of RL-fine-tuned models, particularly under PPO, GRPO, and DPO.

**Strengths:**

1. The paper tackles how RL fine-tuning affects the internal circuitry of LLMs, which is underexplored.
2. The findings are consistent across multiple model families and datasets, offering an interesting mechanistic view of RL’s role in enhancing information flow diversity.

**Weaknesses:**

1. While the paper observes consistent tendency over the three metrices (Activation Intensity, Information Complexity, and Distribution Kurtosis) between SFT and SFT with RL, it does not further verify whether these internal changes causally contribute to downstream performance improvements compared to SFT.

2. The experiments and analyses are limited to mathematical reasoning tasks, raising concerns about the generalizability of the findings to other domains (e.g., natural language understanding, dialogue).

3. All experiments are conducted on 7B-parameter models; including smaller (e.g., 1B) and larger (e.g., 14B) models would strengthen the claim of generality across scales.

**Questions:**

1. Which dataset is used for the visualizations shown in Figure 3?
2. In Table 1, how was the hyperparameter (e.g., truncation scale $\alpha$) chosen?

---

> ### Author Response · Authors · 2025-11-27
> **Rebuttal (Part 1/3)**
>
> We sincerely appreciate your detailed review of this manuscript and the valuable suggestions you have provided. These have been extremely helpful in further improving the paper. Below, we address each of the issues you raised one by one.
>
> ## W1
>
> We appreciate this constructive critique. We agree that establishing a causal link rather than merely a correlation between the observed internal dynamics and downstream performance is essential for the validity of our claims. To address this, we have designed and executed additional experiments to further investigate the relationship between these internal patterns and performance, providing stronger evidence beyond simple correlation, and the relevant code updates have been pushed to the repository.
>
> For this investigation, we employed Qwen2.5-3B-Instruct [1] as the starting point for training and constrained its maximum generation length to 200 tokens, thereby intentionally limiting its ability on mathematical reasoning tasks. Training was performed on the training split of the GSM8K dataset using an improved variant of the GRPO algorithm [2] with a batch size of 32 and 4 samples per query. We utilized a verifiable reward function (RLVR), assigning a score of +3 for accurate solutions, -0.5 for completely incorrect ones, and proportional scores for partial correctness based on prediction deviation.
>
> Our objective was to test the hypothesis that the observed activation patterns are drivers of performance improvements. To do this, we sought to manipulate the three key internal indicators, Activation Intensity, Information Complexity, and Distribution Kurtosis, during the training process. We identified that modulating the sampling temperature serves as an effective mechanism to intervene on these internal states. Specifically, we examined whether inhibiting the "favorable" progression of these metrics (by increasing temperature) would result in diminished capability gains. We monitored the trajectory of these three metrics across the training timeline under two distinct temperature regimes: 0.6 and 1.0. The values reported below represent the deviation from the initial SFT baseline, averaged across $\alpha \in \\{0.1, 0.3, 0.5\\}$. **For a comprehensive visualization of these dynamics, we refer the reviewer to the updated Appendix B.** In the data presentation, we emphasize the peak shifts in the expected direction during the critical mid training phase; values moving further in the expected direction are shown in bold, while the corresponding metrics in the comparison setting are italicized.
>
> ### Activation Intensity ($\uparrow$)
>
> | training steps  | initial |   20    |   40    |   60    |   80    |   100   |    120     |   140    |   160   |
> | :-------------: | :-----: | :-----: | :-----: | :-----: | :-----: | :-----: | :--------: | :------: | :-----: |
> | temperature=0.6 |    0    | -2.2e-6 | 1.9e-5  | 9.5e-5  | 3.1e-4  | 4.0e-4  | **7.6e-4** |  5.7e-4  | 5.8e-4  |
> |    $\Delta$     |         | -2.2e-6 | +2.1e-5 | +7.6e-5 | +2.2e-4 | +8.9e-5 |  +3.6e-4   | -1.9e-4  | +1.2e-5 |
> | temperature=1.0 |    0    | -2.4e-6 | 7.5e-6  | 4.7e-5  | 1.9e-4  | 3.1e-4  |   3.4e-4   | *3.8e-4* | 3.6e-4  |
> |    $\Delta$     |         | -2.4e-6 | +9.9e-6 | +4.0e-5 | +1.4e-4 | +1.3e-4 |  +2.3e-5   | +4.3e-5  | -1.7e-5 |
>
> ### Distribution Kurtosis ($\downarrow$)
>
> | training steps  | initial |   20    |   40    |   60    |   80    |   100   |     120     |    140    |   160   |
> | :-------------: | :-----: | :-----: | :-----: | :-----: | :-----: | :-----: | :---------: | :-------: | :-----: |
> | temperature=0.6 |    0    | -5.4e+0 | -1.6e+1 | -3.8e+1 | -6.1e+1 | -9.3e+1 | **-1.0e+2** |  -9.9e+1  | -9.3e+1 |
> |    $\Delta$     |         | -5.4e+0 | -1.0e+1 | -2.2e+1 | -2.3e+1 | -3.2e+1 |   -1.1e+1   |  +5.1e+0  | +5.9e+0 |
> | temperature=1.0 |    0    | -2.1e+1 | -2.6e+1 | -4.9e+1 | -6.9e+1 | -7.8e+1 |   -7.1e+1   | *-8.1e+1* | -7.0e+1 |
> |    $\Delta$     |         | -2.1e+1 | -4.6e+0 | -2.3e+1 | -2.1e+1 | -8.2e+0 |   +6.9e+0   |  -1.1e+1  | +1.1e+1 |
>
> ### Information Complexity ($\uparrow$)
>
> | training steps  | initial |   20    |   40    |   60    |    80    |   100   |    120     |   140   |   160   |
> | :-------------: | :-----: | :-----: | :-----: | :-----: | :------: | :-----: | :--------: | :-----: | :-----: |
> | temperature=0.6 |    0    | 7.3e-2  | 8.0e-3  | -4.9e-2 |  2.6e-2  | 4.9e-2  | **1.6e-1** | 1.4e-1  | 8.8e-2  |
> |    $\Delta$     |         | +7.3e-2 | -6.5e-2 | -5.7e-2 | +7.6e-2  | +2.3e-2 |  +1.1e-1   | -2.2e-2 | -5.4e-2 |
> | temperature=1.0 |    0    | 1.2e-1  | 1.5e-2  | 3.1e-2  | *5.9e-2* | 5.8e-2  |   4.5e-2   | -2.4e-2 | -3.9e-2 |
> |    $\Delta$     |         | +1.2e-1 | -1.0e-1 | +1.6e-2 | +2.8e-2  | -1.4e-3 |  -1.3e-2   | -6.8e-2 | -1.6e-2 |

---

> ### Author Response · Authors · 2025-11-27
> **Rebuttal (Part 2/3)**
>
> The data reveals that during the intermediate training phase (specifically steps 40–100 for “temperature=1.0” setting and 60–120 for “temperature=0.6”), all three tracked metrics exhibit trends consistent with our hypothesis. It is well-established that elevated sampling temperatures induce greater stochasticity in LLM outputs [3] [4]. This theoretical expectation is corroborated by our empirical results in Figure 4(b) (updated Appendix B), where the standard deviation of group rewards is noticeably higher in the high-temperature setting. We suggest that this high variability makes it difficult for the model to identify correct solution patterns. This prevents the formation of the stable internal pathways needed for mathematical reasoning. As a result, the high-temperature setting restricts the expected internal changes: the peaks of Activation Intensity and Information Complexity are much lower, while Distribution Kurtosis stays higher compared to the low-temperature baseline. Finally, we measured the GSM8K test accuracy (%) for both settings to see the impact.
>
> ### Accuracy ($\uparrow$)
>
> | training steps  | initial |  20  |  40  |  60  |  80  | 100  | 120  | 140  |   160    |
> | :-------------: | :-----: | :--: | :--: | :--: | :--: | :--: | :--: | :--: | :------: |
> | temperature=0.6 |  18.0   | 17.1 | 18.7 | 22.4 | 31.8 | 40.7 | 48.7 | 49.8 | **51.2** |
> |    $\Delta$     |         | -0.9 | +1.6 | +3.7 | +9.4 | +8.9 | +8.0 | +1.1 |   +1.4   |
> | temperature=1.0 |  17.7   | 17.6 | 18.7 | 22.6 | 27.0 | 32.4 | 34.6 | 36.3 |  *36.5*  |
> |    $\Delta$     |         | -0.1 | +1.1 | +3.9 | +4.4 | +5.4 | +2.2 | +1.7 |   +0.2   |
>
> The results show that the biggest performance gains happen in the middle of training (around steps 40–100 for temperature 1.0 and 60–120 for temperature 0.6). This matches the time period when the three metrics change in the expected way. Specifically, the model trained with temperature 0.6 improves much faster and reaches a much higher final accuracy. In contrast, the model trained with temperature 1.0 performs much worse. We believe this supports our causal hypothesis empirically: when we block the formation of these internal patterns (by using high temperature), the model cannot improve as much. This suggests that these patterns are likely needed for successful RL training.
>
>
>
> [1] https://huggingface.co/unsloth/Qwen2.5-3B-Instruct.
>
> [2] Yu Q, Zhang Z, Zhu R, et al. Dapo: An open-source llm reinforcement learning system at scale[J]. arXiv preprint arXiv:2503.14476, 2025.
>
> [3] Liu J, Gao F, Wei B, et al. What can rl bring to vla generalization? an empirical study[J]. arXiv preprint arXiv:2505.19789, 2025.
>
> [4] Renze M. The effect of sampling temperature on problem solving in large language models[C]//Findings of the association for computational linguistics: EMNLP 2024. 2024: 7346-7356.
>
> ----
>
> ## W2
>
> We appreciate the comment regarding the applicability of our findings to domains like NLU and dialogue. We chose to focus on mathematical reasoning primarily because it offers clear, objective standards for correctness. In contrast, tasks like dialogue often lack such rigid benchmarks; improvements in these areas can be difficult to distinguish from simple changes in style, such as the model becoming more verbose or polite [1]. Furthermore, mathematical problems typically demand convergent logical steps to reach a correct solution. This provides a stable environment for tracking internal changes. On the other hand, open-ended NLU or conversational tasks allow for a wide variety of valid responses. This variability creates significant noise in the analysis, making it hard to determine if changes in the model's internal weights are due to improved reasoning or simply the change of the generated text.
>
> Despite this specific focus, the overall consistency of our results across various models, datasets, and algorithms suggests that the expansion of internal pathways is likely a general feature of the online RL process.
>
> To ensure our conclusions are scientifically rigorous, we will revise the manuscript to explicitly limit our primary claims to the domain of mathematical reasoning. We will also update the Limitations section to acknowledge that verifying our findings in open-ended domains like dialogue remains an important direction for future work.
>
> [1] Caglayan O, Madhyastha P S, Specia L. Curious case of language generation evaluation metrics: A cautionary tale[C]//Proceedings of the 28th International Conference on Computational Linguistics. 2020: 2322-2328.

---

> ### Author Response · Authors · 2025-11-27
> **Rebuttal (Part 3/3)**
>
> ## W3
>
> We appreciate your suggestion and fully agree that checking different model sizes is important. However, our experiments focused on 7B models mainly due to GPU memory limitations. Analyzing internal circuit metrics requires memory that is several times larger than the model itself, so the 7B model is the largest size we can handle with our current resources. That being said, the 7B scale remains a widely accepted standard for both local deployment and experimental research [1] [2] [3]. Previous studies have also shown that models of this size are capable of achieving strong performance on mathematical reasoning tasks [4].
>
> As for smaller models, please refer to the experiments mentioned in **our response to W1**. The results show that the 3B model changes during training in the way we expected, giving us good reason to believe our conclusions apply to smaller models as well. We acknowledge the limitation of not being able to test larger models, and we have added a clear statement about this in the Limitations section of the revised paper.
>
>
>
> [1] Zhang Y, Li B, Giannakis G B. RefLoRA: Refactored Low-Rank Adaptation for Efficient Fine-Tuning of Large Models[J]. arXiv preprint arXiv:2505.18877, 2025.
>
> [2] An S, Ma Z, Lin Z, et al. Make your llm fully utilize the context[J]. Advances in Neural Information Processing Systems, 2024, 37: 62160-62188.
>
> [3] Feucht S, Atkinson D, Wallace B C, et al. Token erasure as a footprint of implicit vocabulary items in LLMs[C]//Proceedings of the 2024 Conference on Empirical Methods in Natural Language Processing. 2024: 9727-9739.
>
> [4] Li C, Wang W, Hu J, et al. Common 7b language models already possess strong math capabilities[J]. arXiv preprint arXiv:2403.04706, 2024.
>
> ----
>
> ## Q1
>
> We thank the reviewer for their meticulous review and for pointing this out. The figure visualizes results across all evaluated datasets. For (b), the data points are arranged sequentially by dataset ([College Math, GSM8K, MATH]), where for each dataset, we iterate through the $\alpha$ values ([0.03, 0.1, 0.3, 0.5]). Due to space constraints, we omitted specific x-axis labels in the figure itself and regrettably missed clarifying this arrangement in the original caption. We have now updated the caption to address this:
>
> > (a) diversity of activation patterns across inference samples, including data from all datasets and $\alpha$ values; (b) entropy of output edge patterns per node. In (b), data points are arranged sequentially by dataset (College Math, GSM8K, MATH), iterating over $\alpha \in \\{0.03, 0.1, 0.3, 0.5\\}$ for each.
>
> ----
>
> ## Q2
>
> Regarding the selection of the 'question filtering' hyperparameters $\beta$, $\gamma$, and $\delta$, our primary objective was to ensure that the response lengths of the baseline and RL-fine-tuned models remained comparable, while simultaneously retaining a sufficient sample size for robust analysis. By analyzing the distribution of response lengths for each LLM across datasets before and after RL fine-tuning, we determined that the settings $\beta=0.5$, $\gamma=1.5$, and $\delta=0.5$ effectively satisfied these criteria.
>
> For the truncation scale $\alpha$, our selection was guided by the maximum capacity of our GPU memory. We aimed to maximize the upper bound of $\alpha$ while ensuring that the majority of experiments remained computationally feasible. We identified $\alpha=0.5$ as the practical maximum (noting that Out-Of-Memory errors occurred in a small subset of model-dataset combinations at this level). The remaining values ($\{0.03, 0.1, 0.3\}$) were selected to provide a relatively uniform distribution up to this limit, thereby ensuring comprehensive coverage of the truncation scale range.

---

### Official Review · Reviewer_arJk · 2025-11-04

**Soundness:** 2
**Presentation:** 3
**Contribution:** 2
**Rating:** 4
**Confidence:** 3

**Summary:**

This paper uses edge attribution patching to analyze how RL fine-tuning changes LLM internals across four 7B model pairs on math tasks. They find online RL methods (PPO, GRPO) consistently increase activation intensity and diversity in internal pathways, suggesting more redundant and flexible information flow. DPO-trained models don't show these patterns, revealing key differences between preference-based and online RL approaches.

**Strengths:**

This paper makes a valuable contribution by systematically bridging mechanistic interpretability with RL post-training across four model families, three benchmarks, and multiple hyperparameters. The core findings are clear and consistent: online RL methods (PPO, GRPO) reliably increase activation intensity and diversity in internal pathways, while DPO behaves fundamentally differently—providing mechanistic support for existing empirical observations about these algorithms. The experimental design is rigorous with sound methodology, the EAP framework is computationally scalable and theoretically grounded, and the paper excels in reproducibility with open-source code and detailed specifications.

**Weaknesses:**

1. No Causal Link to Performance:
The paper establishes correlation without causation. While the authors demonstrate that RL training correlates with both altered activation patterns and improved performance, they fail to prove that these internal changes actually cause the capability improvements. Critical intervention experiments are missing: for instance, artificially inducing these activation patterns in SFT models to test whether performance improves, or constraining RL training to maintain low activation intensity to assess whether performance degrades. Without such causal validation, the observed patterns may be mere byproducts rather than drivers of RL's effectiveness.
2. Descriptive Rather Than Explanatory:
The paper describes what changes—higher activation intensity and greater diversity—but not why or how RL objectives produce these patterns. It essentially concludes "RL makes models different internally" without explaining the underlying mechanism. Critically, the work lacks any connection to RL theory: there is no discussion of how exploration, credit assignment, policy gradients, or reward shaping might drive these activation changes. This leaves the findings as empirical observations without mechanistic grounding.
3. Proposed Enhancement:
Incorporating RLVR (RL with Verifiable Rewards) would substantially strengthen the paper. RLVR would complete the methodological space—contrasting online RL with verifiable rewards against learned rewards (PPO/GRPO) and offline preferences (DPO)—thereby testing whether the key distinction is truly online versus offline training. For mathematical reasoning tasks, RLVR provides cleaner binary reward signals that could yield more interpretable activation patterns. Importantly, relevant RLVR-trained models already exist (e.g., Qwen2.5-Math) and would be straightforward to analyze within the existing framework.
Combined with additional mechanistic experiments—tracking activation changes throughout training trajectories, ablating specific RL components, and formally connecting observed patterns to RL theory—this approach would transform the paper from a descriptive observation into an explanatory contribution.

**Questions:**

1. From part 2.1, H^(2ℓ) is used to mean two different things?
At the beginning of the paragraph: "Let H^(2ℓ) ∈ R^(B×P×d_model) denote the input hidden state to the ℓ-th layer"
In the next sentence: "The output of the ℓ-th layer, H^(2ℓ), is computed..." Does the same notation represent both the input and output of layer ℓ?

2. Have you analyzed the dynamics of these changes during RL training? At what point in training do activation intensity and diversity increase—early, late, or gradually throughout? Does this correlate with capability improvements?

3. All experiments focus on mathematical reasoning. Have you tested whether these patterns hold for other domains where RL is commonly applied (e.g., coding, creative writing, instruction following)? If not, how confident are you that these are general properties of RL fine-tuning rather than math-specific phenomena?

---

> ### Author Response · Authors · 2025-11-27
> **Rebuttal (Part 1/5)**
>
> Thank you very much for your thorough and insightful review. Your valuable comments are crucial to improving the quality of this work. Below we provide a point-by-point response to each of your suggestions and questions.
>
> ## W1
>
> Thank you very much for your valuable feedback. We fully agree with your perspective that adding experiments with stronger causal implications would enhance the practical significance of our conclusions. In response, we conducted additional experiments to further investigate the relationship between these internal patterns and performance, providing stronger evidence beyond simple correlation. The corresponding updates have already been incorporated into the code repository.
>
> We used Qwen2.5-3B-Instruct [1] as the starting point for training and constrained its maximum generation length to 200 tokens, thereby intentionally limiting its ability on mathematical reasoning tasks. Building on this setup, we conducted experiments on the training split of the GSM8K dataset using an improved variant of the GRPO algorithm [2] for RL training. The batch size was set to 32, and we sampled 4 candidate responses per question. Rewards were provided via the RLVR method: a correct answer received a reward of 3, a completely incorrect answer received −0.5, and intermediate values were assigned proportionally based on the deviation between the model’s prediction and the ground truth.
>
> To test the hypothesis that these activation patterns drive performance improvements, we aimed to control three key quantities during training, Activation Intensity, Information Complexity, and Distribution Kurtosis, and examine whether suppressing their progression in the favorable direction would weaken the final post-training performance. We found that appropriately adjusting the sampling temperature during RL training provides an effective mechanism for modulating these internal indicators.
>
> We tracked the evolution of the three metrics throughout training under two temperature settings, 0.6 and 1.0. For comparison, we report the differences between the RL-trained models (at various training steps) and the initial SFT model, and we average these differences over $\alpha \in \\{0.1, 0.3, 0.5\\}$. **We recommend consulting Appendix B in the updated PDF, where we provide the full trajectories of these metrics during training for a more intuitive understanding of the observed phenomena.** For each temperature, we highlight the extrema of the three metrics in the expected directions during the mid training stages. Values that move further in the expected direction are shown in bold, while the corresponding values under the other temperature setting are italicized.
>
> ### Activation Intensity ($\uparrow$)
>
> | training steps  | initial |   20    |   40    |   60    |   80    |   100   |    120     |   140    |   160   |
> | :-------------: | :-----: | :-----: | :-----: | :-----: | :-----: | :-----: | :--------: | :------: | :-----: |
> | temperature=0.6 |    0    | -2.2e-6 | 1.9e-5  | 9.5e-5  | 3.1e-4  | 4.0e-4  | **7.6e-4** |  5.7e-4  | 5.8e-4  |
> |    $\Delta$     |         | -2.2e-6 | +2.1e-5 | +7.6e-5 | +2.2e-4 | +8.9e-5 |  +3.6e-4   | -1.9e-4  | +1.2e-5 |
> | temperature=1.0 |    0    | -2.4e-6 | 7.5e-6  | 4.7e-5  | 1.9e-4  | 3.1e-4  |   3.4e-4   | *3.8e-4* | 3.6e-4  |
> |    $\Delta$     |         | -2.4e-6 | +9.9e-6 | +4.0e-5 | +1.4e-4 | +1.3e-4 |  +2.3e-5   | +4.3e-5  | -1.7e-5 |
>
> ### Distribution Kurtosis ($\downarrow$)
>
> | training steps  | initial |   20    |   40    |   60    |   80    |   100   |     120     |    140    |   160   |
> | :-------------: | :-----: | :-----: | :-----: | :-----: | :-----: | :-----: | :---------: | :-------: | :-----: |
> | temperature=0.6 |    0    | -5.4e+0 | -1.6e+1 | -3.8e+1 | -6.1e+1 | -9.3e+1 | **-1.0e+2** |  -9.9e+1  | -9.3e+1 |
> |    $\Delta$     |         | -5.4e+0 | -1.0e+1 | -2.2e+1 | -2.3e+1 | -3.2e+1 |   -1.1e+1   |  +5.1e+0  | +5.9e+0 |
> | temperature=1.0 |    0    | -2.1e+1 | -2.6e+1 | -4.9e+1 | -6.9e+1 | -7.8e+1 |   -7.1e+1   | *-8.1e+1* | -7.0e+1 |
> |    $\Delta$     |         | -2.1e+1 | -4.6e+0 | -2.3e+1 | -2.1e+1 | -8.2e+0 |   +6.9e+0   |  -1.1e+1  | +1.1e+1 |
>
> ### Information Complexity ($\uparrow$)
>
> | training steps  | initial |   20    |   40    |   60    |    80    |   100   |    120     |   140   |   160   |
> | :-------------: | :-----: | :-----: | :-----: | :-----: | :------: | :-----: | :--------: | :-----: | :-----: |
> | temperature=0.6 |    0    | 7.3e-2  | 8.0e-3  | -4.9e-2 |  2.6e-2  | 4.9e-2  | **1.6e-1** | 1.4e-1  | 8.8e-2  |
> |    $\Delta$     |         | +7.3e-2 | -6.5e-2 | -5.7e-2 | +7.6e-2  | +2.3e-2 |  +1.1e-1   | -2.2e-2 | -5.4e-2 |
> | temperature=1.0 |    0    | 1.2e-1  | 1.5e-2  | 3.1e-2  | *5.9e-2* | 5.8e-2  |   4.5e-2   | -2.4e-2 | -3.9e-2 |
> |    $\Delta$     |         | +1.2e-1 | -1.0e-1 | +1.6e-2 | +2.8e-2  | -1.4e-3 |  -1.3e-2   | -6.8e-2 | -1.6e-2 |

---

> ### Author Response · Authors · 2025-11-27
> **Rebuttal (Part 2/5)**
>
> It can be observed that, in the mid-stage of training (approximately steps 40–100 for temperature 1.0 and steps 60–120 for temperature 0.6), all three metrics generally follow the trends predicted by our conclusions. It is worth noting that higher sampling temperatures cause LLMs to produce responses with greater mode variability [3] [4]. This aligns with the group-reward standard deviation curves presented in Figure 4(b) (updated Appendix B), where the group-reward std is consistently higher under the higher-temperature setting. We believe this increased variability makes it more difficult for the model to reliably identify the underlying patterns that correspond to correct solutions. As a result, the model becomes less able to activate stable, beneficial internal pathways associated with accurate mathematical reasoning. Consequently, at higher temperatures, the peak values of Activation Intensity and Information Complexity during training are markedly lower, while the minimum value of Distribution Kurtosis is higher, indicating that a higher temperature effectively restricts these metrics throughout RL training.
>
> We then evaluated the GSM8K test performance (measured by accuracy (%)) of models trained for the corresponding number of steps under the two temperature settings.
>
> ### Accuracy ($\uparrow$)
>
> | training steps  | initial |  20  |  40  |  60  |  80  | 100  | 120  | 140  |   160    |
> | :-------------: | :-----: | :--: | :--: | :--: | :--: | :--: | :--: | :--: | :------: |
> | temperature=0.6 |  18.0   | 17.1 | 18.7 | 22.4 | 31.8 | 40.7 | 48.7 | 49.8 | **51.2** |
> |    $\Delta$     |         | -0.9 | +1.6 | +3.7 | +9.4 | +8.9 | +8.0 | +1.1 |   +1.4   |
> | temperature=1.0 |  17.7   | 17.6 | 18.7 | 22.6 | 27.0 | 32.4 | 34.6 | 36.3 |  *36.5*  |
> |    $\Delta$     |         | -0.1 | +1.1 | +3.9 | +4.4 | +5.4 | +2.2 | +1.7 |   +0.2   |
>
> It can be observed that the stage in which performance gains become most pronounced also lies in the mid-phase of training (approximately steps 40–100 for temperature = 1.0 and 60–120 for temperature = 0.6). This aligns closely with the interval in which the three metrics previously discussed exhibit movement in the expected directions. Notably, the experiment conducted with temperature = 0.6 yields substantially larger performance improvements during training and ultimately converges to a significantly higher accuracy. In contrast, the model trained with temperature = 1.0 achieves markedly lower performance. We view this as empirical support for the causal hypothesis: when the formation of these specific internal patterns is suppressed (via high-temperature sampling), the model fails to achieve comparable performance gains. This suggests that the emergence of such activation patterns is likely a necessary component of effective RL fine-tuning.
>
> [1] https://huggingface.co/unsloth/Qwen2.5-3B-Instruct.
>
> [2] Yu Q, Zhang Z, Zhu R, et al. Dapo: An open-source llm reinforcement learning system at scale[J]. arXiv preprint arXiv:2503.14476, 2025.
>
> [3] Liu J, Gao F, Wei B, et al. What can rl bring to vla generalization? an empirical study[J]. arXiv preprint arXiv:2505.19789, 2025.
>
> [4] Renze M. The effect of sampling temperature on problem solving in large language models[C]//Findings of the association for computational linguistics: EMNLP 2024. 2024: 7346-7356.
>
> ----
> ## W2
>
> We sincerely appreciate your valuable suggestions. In Section 4.3, ‘Results and Analysis,’ we did provide explanations for the observed phenomena. However, due to space limitations, these discussions were not elaborated in detail. Based on your comments and some updated experimental results, we have now further enriched this section. You can refer to **Section 4.3 in the revised PDF**, and for your convenience, we have also included the corresponding content here.
>
> > > The gradient update of any post-training method can be expressed in a unified form as follows (Equation (4)) [1]:
> > >
> > > $\nabla_{\theta} \mathcal{J}_\mathcal{A}(\theta) = \mathbb{E}\_{{(q,o)\sim\mathcal{D}}}[ \frac{1}{|o|}\sum\_{t=1}^{|o|} GC\_\mathcal{A}(q,o,t,\pi\_{\text{rd}},\pi\_{\text{ref}},\pi\_\theta)\nabla\_{\theta}\log \pi\_{\theta}(o\_t \mid q,o\_{<t}) ]$
> > >
> > > In this equation, $\mathcal{D}$ denotes the data source, $GC_{\mathcal{A}}$ represents the gradient coefficient derived from reward signals or evaluation rules, $\pi_{\mathrm{rd}}$ denotes the reward model, $\pi_{\mathrm{ref}}$ denotes the reference policy, $\pi_\theta$ denotes the model being trained, $q$ is the query, $o$ is the response generated by the LLM, and $t$ indexes the token position.
> >
> > Based on Equation (4), we can interpret the observed phenomena by analyzing the fundamental differences in the support of the sampling distribution $\mathcal{D}$ and the properties of the gradient coefficient $GC_{\mathcal{A}}$ across SFT, Online RL, and DPO.

---

> ### Author Response · Authors · 2025-11-27
> **Rebuttal (Part 3/5)**
>
> > For SFT, the data source is static, drawn from a fixed human-annotated distribution $\mathcal{D}\_{SFT} = \\{(q, o) \sim P\_{\text{data}}\\}$, with a constant gradient coefficient $GC\_{SFT}=1$. Consequently, the model optimizes its internal representations to minimize cross-entropy on a narrow, predefined manifold of "correct solutions." This drives the model to converge towards a low-entropy mode that mimics the training data, resulting in activations concentrated on a small number of outlier edges (high Distribution Kurtosis) and limited engagement of redundant pathways (low Activation Intensity).
> >
> > In contrast, Online RL algorithms like PPO and GRPO fundamentally alter the data source by introducing on-policy sampling, where outputs are dynamically generated by the evolving policy itself: $\mathcal{D}\_{RL} = \\{(q, \\{o_i\\}\_{i=1}^G) \mid q \sim P\_{\text{data}}, o\_i \sim \pi_\theta(\cdot\mid q) \\}$. This mechanism significantly expands the stochastic support set of the training distribution beyond the SFT subspace, providing the LLM with a richer set of reasoning path samples for each query $q$. Mechanistically, to handle the expanded state space encountered during exploration, the network is compelled to activate and reinforce latent or "dormant" internal circuits that were underutilized during SFT. Furthermore, the gradient coefficient in online RL varies dynamically based on feedback from the reward model or rule. Taking GRPO as an example, $GC_{GRPO}=\hat A_{i,t}(q,o,t,\pi_{rd})+\beta\left(\frac{\pi_{ref}(o_{i,t}|o_{i,<t})}{\pi_\theta(o_{i,t}|o_{i,<t})}-1\right)$. To maximize expected reward, the model is driven to mobilize these less active internal circuits to master relatively "harder" problems, as correct responses to such instances typically yield significantly higher gradient coefficients. The observed increase in Activation Intensity and the simultaneous decrease in Distribution Kurtosis reflect this broader utilization of residual pathways. Moreover, as multiple distinct reasoning paths for the same question are reinforced, the entropy of the internal edge weight distribution increases.
> >
> > Furthermore, from this unified perspective, we can elucidate why DPO exhibits distinct behaviors, particularly its failure to consistently enhance activation intensity and information complexity. Although DPO is mathematically derived from the RL objective, it operates as an offline (or semi-offline, where datasets are refreshed only periodically) algorithm. Its data source remains closer to a relatively more static distribution: $\mathcal{D}\_{DPO} = \\{(q, o^+, o^-) \sim P\_{\text{data}}\\}$, rather than the real-time policy $\pi\_\theta$. Since DPO restricts optimization to the fixed support set of an offline dataset and effectively retains only two potentially stale contrasting samples for each query $q$, the mechanistic pressure to expand the network's functional capacity through stochastic sampling is significantly weaker. This explains why Activation Intensity and Information Complexity do not show a consistent upward trend compared to the SFT baseline. However, DPO does successfully reduce Distribution Kurtosis. This is because the preference optimization objective is driven by the gradient coefficient $GC_{DPO} = \sigma\left(\beta \log \frac{\pi_\theta(o^-|q)}{\pi_{ref}(o^-|q)} - \beta \log \frac{\pi_\theta(o^+|q)}{\pi_{ref}(o^+|q)}\right)$. This soft margin mechanism relaxes the strict token-matching constraints of SFT, favoring a broader reward maximization landscape and thereby inhibiting the emergence of high-intensity activation edges to some extent, which can be intuitively understood as mitigating rote memorization. Thus, while DPO attenuates the model's reliance on a few high-intensity edges during inference (low kurtosis), it lacks the on-policy exploration dynamics inherent to Online RL, which are essential for driving the systematic enhancement of average internal activation intensity and diversity.
>
> In summary, in line with widely observed empirical trends, one of the primary factors driving differences across training methods is whether the support set consists of online-sampled data and how such sampling is conducted [2]. Our newly added experiments largely corroborate this explanation: differences in training temperature effectively induce variations in the sampled data during training, which in turn lead to changes in the model's internal properties and, ultimately, to differences in the LLM's external performance.
>
>
>
> [1] Shao Z, Wang P, Zhu Q, et al. Deepseekmath: Pushing the limits of mathematical reasoning in open language models[J]. arXiv preprint arXiv:2402.03300, 2024.
>
> [2] Sachdeva N, Coleman B, Kang W C, et al. How to train data-efficient llms[J]. arXiv preprint arXiv:2402.09668, 2024.

---

> ### Author Response · Authors · 2025-11-27
> **Rebuttal (Part 4/5)**
>
> ## W3
>
> Thank you for this insightful suggestion. We appreciate the reviewer's intent to complete the methodological space by contrasting online RL with verifiable rewards against learned rewards.
>
> **Clarification on Taxonomy:** We would like to respectfully clarify that RLVR (Reinforcement Learning with Verifiable Rewards) represents a source of reward signal (rule-based verification) rather than a distinct optimization algorithm parallel to PPO or GRPO. In the literature, RLVR is typically implemented using algorithms like PPO or GRPO. For instance, [1] utilizes PPO with verifiable rewards, while [2] employs GRPO with verifiable rewards. Therefore, comparing "RLVR" against "PPO/GRPO" presents a categorical mismatch, as they are orthogonal components of the training pipeline.
>
> Our current experimental suite already covers the spectrum of reward sources and optimization methods the reviewer is interested in:
>
> - Mistral [3]: PPO + Learned Reward Model.
> - Distilled-Qwen [4]: GRPO + **Rule-based Reward (This is effectively RLVR)**.
> - Deepseek-Math [5]: GRPO + Learned Reward Model.
> - Qwen2.5 [6]: DPO + **Rule-based Verification**.
>
> We hope this clarification addresses your concern regarding the completeness of our baselines.
>
>
>
> [1] Lambert N, Morrison J, Pyatkin V, et al. Tulu 3: Pushing frontiers in open language model post-training[J]. arXiv preprint arXiv:2411.15124, 2024.
>
> [2] Mroueh Y. Reinforcement Learning with Verifiable Rewards: GRPO's Effective Loss, Dynamics, and Success Amplification[J]. arXiv preprint arXiv:2503.06639, 2025.
>
> [3] Peiyi Wang, Lei Li, Zhihong Shao, RX Xu, Damai Dai, Yifei Li, Deli Chen, Yu Wu, and Zhifang Sui. Math-shepherd: Verify and reinforce llms step-by-step without human annotations. arXiv preprint arXiv:2312.08935, 2023.
>
> [4] Yang Chen, Zhuolin Yang, Zihan Liu, Chankyu Lee, Peng Xu, Mohammad Shoeybi, Bryan Catanzaro, and Wei Ping. Acereason-nemotron: Advancing math and code reasoning through reinforcement learning. arXiv preprint arXiv:2505.16400, 2025.
>
> [5] Zhihong Shao, Peiyi Wang, Qihao Zhu, Runxin Xu, Junxiao Song, Xiao Bi, Haowei Zhang, Mingchuan Zhang, YK Li, Yang Wu, et al. Deepseekmath: Pushing the limits of mathematical reasoning in open language models. arXiv preprint arXiv:2402.03300, 2024.
>
> [6] Hanning Zhang, Jiarui Yao, Chenlu Ye, Wei Xiong, and Tong Zhang. Online-dpo-r1: Unlocking effective reasoning without the ppo overhead, 2025. Notion Blog, 2025a.
>
> ----
>
> ## Q1
>
> We sincerely appreciate your careful review. This was a minor typo, which we have now corrected to
>
> > Let $\mathbf{H}^{(2\ell)} \in \mathbb{R}^{B \times P \times d_{\text{model}}}$ denote the input hidden state to the $(\ell+1)$-th layer, where $B$ is the batch size ...
>
> ----
>
> ## Q2
>
> Thank you for this excellent question regarding the temporal dynamics of training.
>
> As detailed in our response to W1 and illustrated in updated Appendix B, we tracked the evolution of Activation Intensity, Information Complexity, and Distribution Kurtosis throughout the training process. We observed that under effective temperature settings, these metrics exhibit significant movement in their expected directions, specifically a decrease in Distribution Kurtosis concurrent with increases in Information Complexity and Activation Intensity, during the mid-stage of training (approximately steps 40–100 for temperature = 1.0 and 60–120 for temperature = 0.6). Crucially, this period perfectly aligns with the phase of most rapid performance improvement on the GSM8K dataset. This temporal synchronization provides strong evidence that the expansion of internal pathways is closely tied to the acquisition of reasoning capabilities.
>
> In addition, we made several interesting observations regarding the dynamics of RL training. As shown in Figure 5 in updated Appendix B, regardless of the temperature setting, model performance continues to increase slightly during the late stage of training (steps 140–160). However, during this same interval, Distribution Kurtosis exhibits a slight upward trend, while Information Complexity and Activation Intensity show mild declines. One possible interpretation is that, during the mid-phase of training, the model acquires richer and more stable beneficial response patterns through feedback from RL sampling. Once most of these new patterns have been internalized, the late-phase training dynamics become more similar to SFT, where the model repeatedly reinforces the patterns it has already learned. As a result, the diversity of internal pathways gradually decreases, and the model’s internal behavior becomes more concentrated.

---

> ### Author Response · Authors · 2025-11-27
> **Rebuttal (Part 5/5)**
>
> ## Q3
>
> We appreciate this suggestion regarding domain generalization. Our decision to concentrate on mathematical reasoning is driven by two primary factors. First, this domain benefits from established research baselines and objective evaluation metrics. In contrast, domains like creative writing lack robust measures to verify whether RL training has genuinely improved capabilities versus merely altering stylistic preferences [1]. Second, mathematical problems typically demand convergent logical steps to reach a correct solution, providing a stable baseline for tracing internal activations. Conversely, in domains like coding or creative writing, RL often incentivizes diverse or structurally complex outputs (e.g., more elaborate code architectures) to maximize reward. This output heterogeneity introduces significant noise, making it difficult to decouple genuine mechanistic shifts in edge weights from variations caused solely by the divergent nature of the generated content.
>
> Nevertheless, given the consistency of our findings across multiple model families, RL algorithms, and datasets, we believe that the observed internal dynamics, specifically the expansion of activation pathways, are likely intrinsic to the online RL process rather than specific to mathematics. The mechanism of reinforcing diverse valid pathways via on-policy sampling is theoretically applicable to any reasoning-intensive task.
>
> However, to maintain strict scientific rigor, we have revised the manuscript to explicitly scope our primary claims to mathematical reasoning. We have also expanded the Limitations section to clarify that while theoretical evidence and consistency suggest broader applicability, empirical verification in domains with open-ended outputs remains a subject for future investigation.
>
>
>
> [1] Caglayan O, Madhyastha P S, Specia L. Curious case of language generation evaluation metrics: A cautionary tale[C]//Proceedings of the 28th International Conference on Computational Linguistics. 2020: 2322-2328.

---

### Meta-Review · Area_Chair_ovje · 2026-01-07

**Summary:**

**The primary concern** among all three reviewers is that the paper is descriptive, rather than explanatory. They all acknowledge that the paper provides strong evidence for differently evolving internal activation statistics and differences in the evolving circuitry within the transformers; however reviewers arJk and VkAD ask if a causal relationship between these internal properties and the post-trained models downstream performance can be established.

**Secondary concerns** mentioned the reviewers revolve around generalizability of the results. Specifically:
1) Do the results hold hold for larger (smaller) models than the 7B models investigated
2) Do these results transfer to other domains beyond mathematical reasoning
3) Using RL with Verifiable Rewards (RLVR) as an additional RL approach

**Reviewer Concerns:**

**Primary Concern**
During the rebuttal the authors present an intervention study on Qwen2.5-3B to address the primary concern regarding causality vs. correlation. Their additional experiments demonstrate that high-temperature sampling inhibits the formation of the observed internal patterns (intensity/diversity) and results in inferior performance, whereas lower temperatures facilitate these patterns and improve accuracy.
I am not convinced that these experiments conclusively answer the causation vs. correlation questions raised by the reviewers. However they provide additional interesting evidence and strengthen the paper significantly.

**Secondary concerns**

1) The authors provide  additional experiments with a 3B model and observe the same quantitative behavior as for the 7B models in the original submission. The authors also argue (IMHO correctly), that 7B models are generally considered reasonably sized systems for this kind of research. I think we should also credit the authors for using 7B models from 4 different model families. I think this concern is sufficiently addressed.
2) The authors clarify and defend that this work indeed focuses on mathematical reasoning tasks. This thus remains an outstanding concern.
3) RLVR/Methodology: The authors clarify that their existing baselines (e.g., Distilled-Qwen using GRPO+Rule-based Reward) effectively represent RLVR, and that RLVR is a reward source rather than a distinct optimization algorithm.

**Reviewer Scores:**

**Reviewer arJk (Current: 4): likely increase to 5.**

This reviewer provided the most specific critique regarding the lack of causal links and the need for RLVR. The authors explicitly ran the intervention experiment (temperature modulation) to address causality and provided a clear rebuttal regarding RLVR taxonomy.

**Reviewer VkAD (Current: 4): likely increase to 5**
This reviewer focused on causality and scale/domain generalization. The inclusion of the 3B model results and the causality experiments addresses the bulk of their technical concerns. The domain limitation however remains.

**Reviewer kREr (Current: 8): likely remain at 8**
The reviewer was generally positive; the additional experiments touch upon the questions and concerns raised by this reviewer, provide additional evidence and strengthen the paper. However they are not a fundamental game-changer for interpreting the results.

---

### Decision · Program_Chairs · 2026-01-26

Accept (Poster)